# Ancient hybridization fuels rapid cichlid fish adaptive radiations

Joana I. Meier[1,2,3], David A. Marques[1,2,3], Salome Mwaiko[1,2], Catherine E. Wagner[1,2,4], Laurent Excoffier[3,5] & Ole Seehausen[1,2]

Understanding why some evolutionary lineages generate exceptionally high species diversity is an important goal in evolutionary biology. Haplochromine cichlid fishes of Africa's Lake Victoria region encompass >700 diverse species that all evolved in the last 150,000 years. How this 'Lake Victoria Region Superflock' could evolve on such rapid timescales is an enduring question. Here, we demonstrate that hybridization between two divergent lineages facilitated this process by providing genetic variation that subsequently became recombined and sorted into many new species. Notably, the hybridization event generated exceptional allelic variation at an opsin gene known to be involved in adaptation and speciation. More generally, differentiation between new species is accentuated around variants that were fixed differences between the parental lineages, and that now appear in many new combinations in the radiation species. We conclude that hybridization between divergent lineages, when coincident with ecological opportunity, may facilitate rapid and extensive adaptive radiation.

[1] Aquatic Ecology and Evolution, Institute of Ecology and Evolution, University of Bern, 3012 Bern, Switzerland. [2] Department of Fish Ecology and Evolution, Centre for Ecology, Evolution & Biogeochemistry, Eawag: Swiss Federal Institute of Aquatic Science and Technology, 6047 Kastanienbaum, Switzerland. [3] Computational and Molecular Population Genetics Lab, Institute of Ecology and Evolution, University of Bern, 3012 Bern, Switzerland. [4] Biodiversity Institute & Department of Botany, University of Wyoming, Laramie Wyoming 82071, USA. [5] Swiss Institute of Bioinformatics, 1015 Lausanne, Switzerland. Correspondence and requests for materials should be addressed to O.S. (email: ole.seehausen@eawag.ch).

Heterogeneity in diversification rates among lineages is a major factor shaping biodiversity. Yet, biological and environmental factors underlying this variation are incompletely understood[1]. Adaptive radiations are prime study systems to learn about these factors as they are characterized by a rapid origin of many species showing a diversity of ecological adaptations. This process requires high levels of heritable variation in traits related to ecological and reproductive isolation. However, adaptive radiations are often too rapid for the emergence of new relevant mutations between successive speciation events[2] and are thus more likely to stem from standing variation. Hybridization between species can instantaneously boost genetic variation, which may facilitate speciation and adaptive radiation[3–9].

Hybridization among members of an adaptive radiation, has been suggested to potentially facilitate speciation events, an idea known as the 'syngameon hypothesis'[3]. Introgression of traits involved in adaptation or reproductive isolation has been demonstrated among members of several adaptive radiations (for example, traits related to host shift in *Rhagoletis* fruit flies[10], wing patterns in *Heliconius* butterflies[11,12] or beak shape in Darwin's finches[13]). In other radiations, the hybrid ancestry of some species has been inferred, but a direct link between introgressed traits and speciation awaits further testing (for example, cichlid fishes of Lakes Tanganyika[14–16], Malawi[17], Victoria[18,19] and Barombi Mbo[20]).

Another hypothesis for a perhaps more fundamental role of hybridization in adaptive radiation, distinct from the 'syngameon hypothesis', is the idea that hybridization between distinct lineages may seed the onset of an entire adaptive radiation[3]. Such hybridization can be common when allopatric lineages come into secondary contact[4], and selection against hybrids may be weak during colonization of new environments. In this situation, the formation of a hybrid swarm, if coincident with ecological opportunity, may accelerate adaptive radiation by (i) providing functional genetic variation that can recombine into novel trait combinations favoured by selection and mate choice, and (ii) breaking genetic correlations that constrained the evolvability of parental lineages[3]. In addition, hybridization may facilitate speciation when multiple fixed differences that confer reproductive isolation between the two parental species decouple and segregate in a hybrid swarm, such that selection against incompatible gene combinations can generate more than two new reproductively isolated species[21–23].

This 'hybrid swarm origin of adaptive radiation' hypothesis has been more challenging to test. So far the only adaptive radiation for which a hybrid origin has been robustly demonstrated is the Hawaiian silverswords, which have radiated from an allopolyploid hybrid population between two North American tarweed species[24]. Because gene and genome duplication are also proposed to facilitate adaptive radiation[25], it is difficult to distinguish between effects of hybridization per se and those of gene or genome duplication in this case. Evidence consistent with a hybrid swarm origin of entire radiations has also been found in Alpine whitefish[26], the 'mbuna' group of the Lake Malawi cichlid fish radiation[27], and allopolyploid Hawaiian endemic mints[28,29] and possibly other polyploid plant radiations on Hawaii[30]. However, it remains to be tested if, in these systems, hybridization occurred before or after the radiation had started, and if hybridization-derived polymorphisms played a role in speciation and adaptive diversification.

The Lake Victoria Region Superflock of cichlid fish (LVRS) is a group of 700 haplochromine cichlid species endemic to the region around Lake Victoria and nearby western rift lakes in East Africa that started diversifying about 100–200 thousand years ago[31–34]. It includes several adaptive radiations, one in each of the major lakes of the region (Lakes Victoria, Edward, Albert and Kivu). The largest of them is in Lake Victoria, which has at least 500 endemic species that evolved in the past 15,000 years[34–36]. Each radiation comprises enormous diversity in habitat occupation, trophic ecology, colouration and behaviour. The high diversification rate but also the high nuclear genomic variation in the LVRS despite its young age[34,37] suggest that large amounts of standing genetic variation must have been present at the onset of the radiation[2,3]. Previous work showing cytonuclear discordance in phylogenetic reconstructions between LVRS and several riverine cichlid species raised the possibility of ancient hybridization between divergent species at the base of the radiation but could not demonstrate it[37]. Hybridization on secondary contact is not unlikely, as allopatric cichlid species, divergent by even millions of years, readily produce fertile offspring in the lab[38].

Using genomic data from riverine haplochromine cichlids sampled from all major African drainage systems, and representative species from all lineages within the Lake Victoria region, we demonstrate here that the LVRS evolved from a hybrid swarm. All lake radiations show very similar proportions of mixed ancestry derived from two distantly related haplochromine lineages that had evolved in isolation from one another in different river systems for more than a million years before hybridizing in the Lake Victoria region. We find evidence that this hybridization event facilitated subsequent adaptive radiation by providing genetic variation that has been recombined and sorted into many new species. Variants that were fixed between the parental lineages show accentuated differentiation between young Lake Victoria species, but appear in many new combinations in the different species. Notably, each of the two major allele classes of an opsin gene involved in adaptation and speciation among Lake Victoria cichlids[39,40] is likely derived from one of the two parental lineages. This indicates that a major part of the variation at this gene segregating in the LVRS stems from hybridization between these lineages. Our results suggest that hybridization between relatively distantly related species, when coincident with ecological opportunity, may facilitate rapid adaptive radiation. Thus, hybridization, even in the distant past, may have important implications for understanding variation in extant species richness between lineages as well as variation in recent rates of diversification.

## Results

**Identifying the closest relatives of the LVRS.** To identify the closest relatives of the LVRS, we use comprehensive sampling of haplochromine cichlids from the LVRS and from all major river systems harbouring haplochromines (Fig. 1, Supplementary Data 1). Maximum likelihood phylogenetic trees were reconstructed from 3.15 Mb of concatenated restriction site associated DNA (RAD) sequences (436,166 SNPs, 8.1% missing data, Supplementary Data 2), and from two mitochondrial markers (1,897 bp). In the nuclear phylogeny, the entire LVRS forms a well-supported clade that also includes cichlids from the lower Nile in Egypt (*Haplochromis* spp. Egypt) and *Haplochromis* sp. 'Nyangara' from the Rusizi River, the outflow of Lake Kivu and an inflow to Lake Tanganyika (orange labels in Fig. 1, Supplementary Fig. 1, Supplementary Discussion). *Astatotilapia stappersi* from the Kalambo River (an inflow to Lake Tanganyika, Congo drainage, referred to as *Haplochromis sp.* 'Chipwa' in Meyer et al.[41]), and an undescribed *Astatotilapia* species from the middle Congo (*A. sp.* 'Yaekama', DRC), together hereafter called the 'Congolese lineage' (red labels), form the superflock's closest relatives. The sister group to the LVRS plus the Congolese lineage taxa is a clade including lineages in

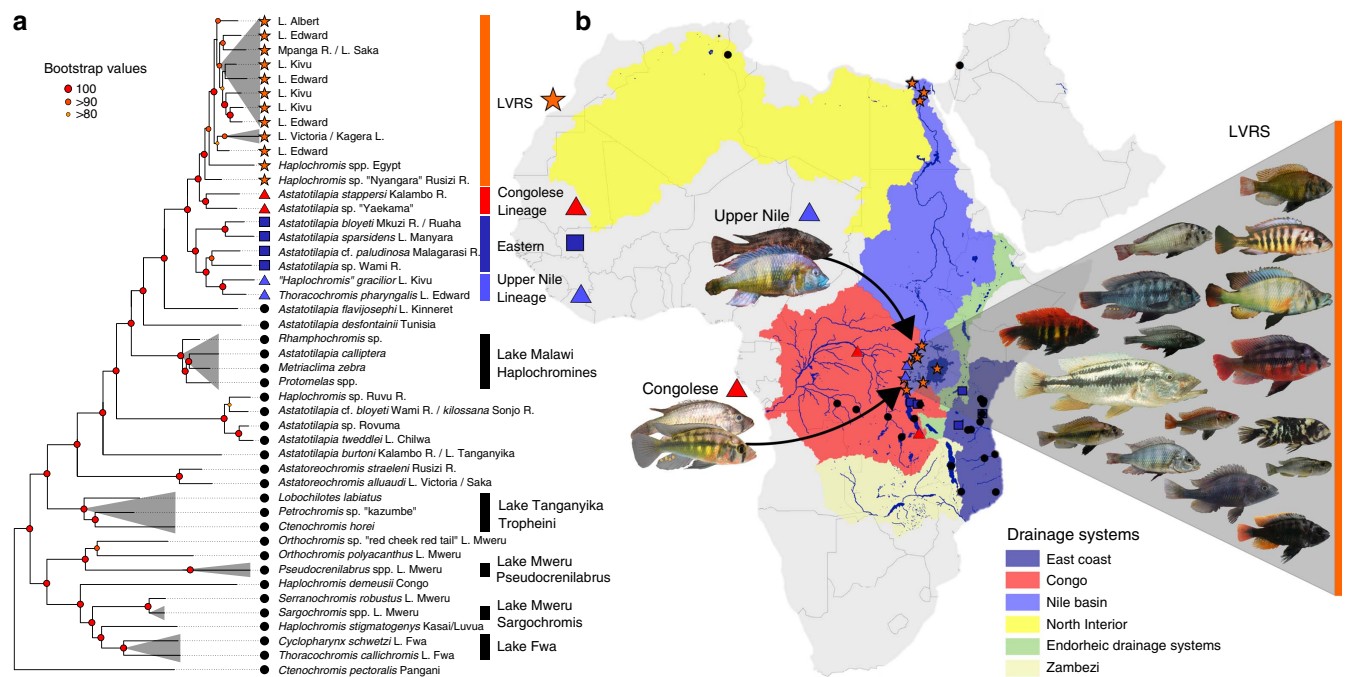

**Figure 1 | Phylogenetic context of the Lake Victoria Region cichlid radiation. (a)** Maximum likelihood phylogeny built from concatenated RADtag sequences of Lake Victoria Region Superflock (LVRS) cichlids and relatives including all known lineages of haplochromine cichlids ($n = 156$). Radiations are indicated as grey triangles in the phylogenetic tree and multiple samples of a lineage are visually collapsed to a single terminal branch (full tree in Supplementary Fig. 1). Members of the LVRS (including *Haplochromis* sp. 'Nyangara' from the Rusizi River and *Haplochromis* spp. Egypt, see Supplementary Discussion) are indicated with orange stars both in the tree and in the sampling map **(b)** and are labelled by lake (L) or river (R) they were sampled in. 'Congolese lineage' LVRS relatives are highlighted with red triangles, members of the 'Upper Nile lineage' with blue triangles, those from Eastern rivers with dark blue squares, and all other more distantly related lineages with black circles. **(b)** Sampling map. River drainage systems that we sampled are shown as coloured polygons. The radiation ancestor's closest living relatives are shown in images: '*Haplochromis*' *gracilior/Thoracochromis pharyngalis* from the Upper Nile lineage and *Astatotilapia* sp. 'Yaekama'/*A. stappersi* from the Congolese lineage. The Lake Victoria cichlids shown in the grey triangle on the right represent some of the many and varied species that arose from the hybrid swarm (Photo credits: Ole Seehausen, Salome Mwaiko, Frans Witte, 'Teleos', Uli Schliewen, Adrian Indermaur, Oliver Selz; map adapted from http://www.worldwildlife.org/hydrosheds[80]).

endorheic (that is, closed) drainage systems of the Eastern Rift and in Indian Ocean drainage systems, east of the Lake Victoria region ('Eastern taxa', dark blue), as well as '*Haplochromis*' *gracilior* from Lake Kivu and *Thoracochromis pharyngalis* from Lake Edward (light blue, Supplementary Discussion). Although the latter two species are sympatric with LVRS members, they are clearly not part of the radiations themselves. We refer to *H. gracilior* and *T. pharyngalis* as the 'Upper Nile lineage' because all known members are confined to the region that was historically the uppermost Nile drainage[37,42]. Consistent with previous publications[33,43], we estimate the split of this lineage from the Congolese lineage to date to ~1.6–5.8 million years ago, coincident with the uplift of the flanks of the Western branch of the East African Rift truncating the paleo-river network and isolating the LVR (the new Upper Nile) from the Congo (Supplementary Table 1; Supplementary Fig. 3).

Our genome-wide nuclear sampling provides unprecedented resolution of the relationships between radiations within the LVRS (Fig. 1, Supplementary Fig. 1). The radiations in Lake Victoria (including its major tributary, the Kagera River), in Crater Lake Saka and associated Mpanga River, and in Lake Albert, each form monophyletic clades. In contrast, the Lake Edward members of the LVRS are a paraphyletic group that includes taxa basal to the radiations of Lakes Victoria and Saka, and others that form a clade together with species from Lake Kivu. These findings are consistent with the hypothesis[34] that Lake Edward constitutes the oldest extant radiation within

the LVRS, from which the radiations in the others lakes are derived, and that connections between Lakes Edward and Kivu existed until recently.

**Tests of hybridization**. Phylogenetic reconstruction based on concatenated sequence data assumes a single history across the genome, although recombination, incomplete lineage sorting, and introgressive hybridisation can cause extensive genome-wide variation in genealogy[44]. To account for such genealogical variation, we reconstructed a SNP based species tree[45]. The resulting tree supports the sister relationship of the LVRS with the Congolese taxa, but reveals incongruence between this group and the Eastern clades and Upper Nile clade (Supplementary Fig. 2). To test for genetic admixture, we computed Patterson's D statistics (ABBA-BABA test)[46]. We found that all LVRS radiations show strong signals of admixture with the two Upper Nile species, *H. gracilior* and *T. pharyngalis* (Fig. 2, Supplementary Table 2, Supplementary Fig. 4). Excess allele sharing between the LVRS and either species is equally strong (Supplementary Table 3, rows 2.1-2.4, Supplementary Fig. 5). Importantly, neither *H. gracilior* nor *T. pharyngalis* show greater admixture with the LVRS species with which they live in sympatry than with allopatric members of the LVRS from other lakes that have been geographically isolated for many thousands of years (Fig. 2 and Supplementary Fig. 5).

To test the directionality of gene flow, we applied an extended version of the partitioned D statistic[47] (5 population test,

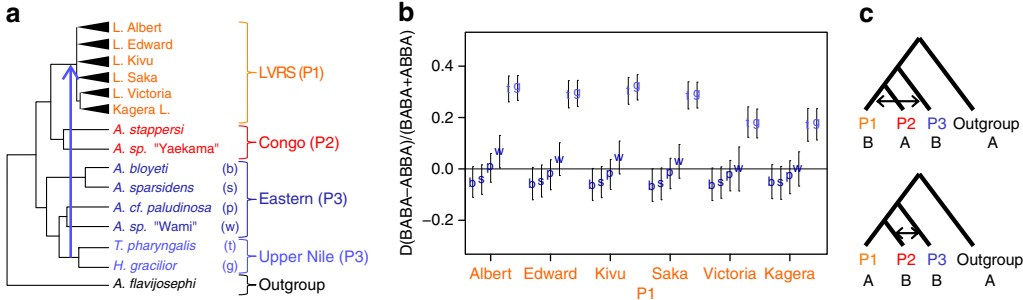

**Figure 2 | Evidence for Congo-Nilotic hybridization in the ancestry of the LVRS.** (**a**) Schematic genealogy with taxa used for D statistics ($n = 73$ individuals, see Supplementary Data 2). Abbreviations used in other panels are given in parentheses and the color scheme is the same as in Figure 1. The inferred gene flow edge is shown with an arrow (Note the directionality of gene flow is inferred with the five-population test not shown in this figure). (**b**) D statistics to test for potential gene flow between each Eastern and Upper Nile taxon (P3) separately (abbreviations given in **a**) and cichlids from each LVR lake radiation (P1) or the Congolese taxon *A. stappersi* (P2). Vertical bars correspond to three standard errors. Positive D values indicate gene flow between P1 (LVR lake radiation) and P3 (Eastern or Upper Nile taxon), whereas negative D values indicate gene flow between P2 (*A. stappersi*) and P3 (Eastern or Upper Nile taxon) as illustrated in (**c**). Exact values and more test results are given in Supplementary Table 2.

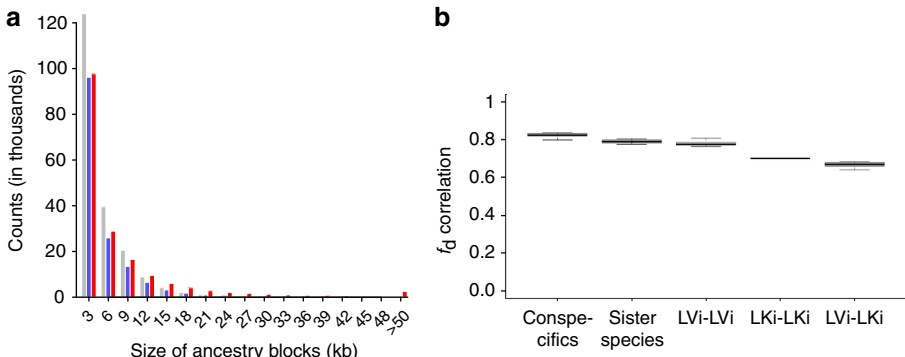

**Figure 3 | Congo-Nilotic ancestry blocks in LVRS genomes.** (**a**) The size distribution of putative ancestry blocks shows mostly small ancestry blocks and slightly larger Congolese (red) than Upper Nile blocks (blue). The plot shows the counts of ancestry blocks in different size categories summed up for five LVRS radiation species across all scaffolds calculated with 3 kb windows. As most blocks do not span multiple windows of 3 kb and many blocks cannot be clearly allocated to Congolese or Upper Nile ancestry (grey) (Supplementary Fig. 8), the average ancestry block size is likely 3 kb or smaller, consistent with hybridization many thousands of years ago. (**b**) Correlation of ancestry blocks between whole-genome sequenced LVRS members is high overall but decreases with phylogenetic distance. The boxplots show correlation of $f_d$ (ref. 78) in 10 kb windows between single individuals of conspecifics (*Pundamilia* individuals of the same species), sister species (*Pu. pundamilia* versus *Pu. nyererei*), more distantly related Lake Victoria (LVi) species (*Paralabidochromis flavus* versus *Pu. pundamilia* and *Pu. nyererei*), Lake Kivu (LKi) species (*Pa. paucidens* versus *Harpagochromis vittatus*) and Lake Victoria against Lake Kivu species. This suggests that all radiation member species share the same hybridization event in their ancient history but vary in how long after that event they remained part of the same recombining population. It also suggests that some of the admixture variation still segregates within individual species (indicated by the deviation from an $f_d$ correlation of 1 among conspecifics).

Supplementary Table 3). We find evidence for gene flow from *H. gracilior* (g in Fig. 2b) and from *T. pharyngalis* (t in Fig. 2b) into each LVRS lake radiation, but no evidence for gene flow in the opposite direction (Supplementary Table 3, rows 1.1–1.4). Using F4-ratio tests[48], we estimated the Upper Nile ancestry proportion to be ∼20% in all LVRS radiations (Supplementary Fig. 6 and Supplementary Table 4). Whole-genome sequencing of Lake Victoria and Lake Kivu radiation species, and *A. stappersi*, *T. pharyngalis* and *H. gracilior*, confirms excess allele sharing between LVRS members and the Upper Nile lineage, and reveals ancestry blocks of ∼3 kb, consistent with evidence that the admixture event occurred many thousands of generations ago in the common ancestor of all superflock radiations in the different lakes (Fig. 3a, Supplementary Fig. 8). The genome-wide signatures of admixture are correlated among different species from Lake Victoria and Lake Kivu (Fig. 3b), in line with a shared admixture event at the onset of the LVRS.

**Importance of the ancient hybridization event.** Many phenotypic traits known to contribute to ecological adaptation and reproductive isolation have diverged in multiple speciation events in each of the LVRS radiations (for example, tooth shapes, male nuptial colouration, opsin alleles)[39,40]. Some of these traits are also divergent between the Congolese *Astatotilapia* and Upper Nile *Thoracochromis* species[37,42]. Intrinsic incompatibilities have also been found in experimental hybrid crosses of similarly divergent *Astatotilapia* species[38]. In addition, some of the allelic variation in the lake radiations has been shown to predate the origin of the LVRS (refs 49,50), perhaps most notably that of the long-wavelength sensitive (*LWS*) opsin gene, one of the best-studied genes in Lake Victoria cichlids[51]. This gene codes for the protein moiety of red-sensitive visual pigments in retinal cones and is exceptionally diverse in Lake Victoria Region cichlids[51]. It plays a crucial role in adaptation on the steep ambient light gradients associated with water depth and turbidity gradients that are characteristic for the lakes in the Lake

Victoria region. More red-shifted *LWS* opsin variants are beneficial in deep and murky water where the light spectrum is relatively more red-shifted because of particulate matter absorbing and scattering light of shorter wave lengths[39,52]. This opsin gene likely also plays a role in behavioural reproductive isolation because divergence in colour perception between species with different *LWS* opsin genotypes[53] is often associated with divergent male breeding colouration[39,49,54], which is an important mate choice cue[55].

The *LWS* opsin gene is highly diverse in the LVRS with two deeply divergent haplotype clades, often referred to as haplotype classes because the alleles within each clade are functionally similar to each other[51]. A substitution at amino acid position 177 in the cichlid *LWS* gene shifts peak absorbance towards longer wavelength (red) in haplotype class I relative to class II by 7 nm (refs 49,51) and also other substitutions influence spectral sensitivity (Supplementary Discussion, Supplementary Data 3). The two haplotype classes are often associated with different light environments, class I with shallow and clear water, and class II with deep and turbid water. For example, rocky shore algae scrapers of the genera *Neochromis* and *Mbipia* and mud bottom detritivores of the genus *Enterochromis*, likely representing an early divergence event in the Lake Victoria radiation, occupy the opposite ends of the light spectrum in Lake Victoria: clear and shallow versus murky and deep. All 438 *LWS* haplotypes previously sequenced from *Neochromis* and *Mbipia* algae scrapers[39,49] are part of the haplotype class I, whereas 11 of the 12 *Enterochromis* *LWS* haplotypes belong to class II (ref. 39). Within ecologically variable lineages, young sister species occupying shallow versus deep or clear versus turbid waters, often have different frequencies of these two *LWS* haplotype classes or recombinants between them, and *LWS* variation is consistently associated with divergence in male nuptial coloration (Fig. 4, Supplementary Discussion). We sequenced the *LWS* opsin gene in Congolese and Upper Nile taxa, and found that the two haplotype classes[51] in the LVRS are each shared exclusively with just one of these parental lineages (Fig. 4, Supplementary Fig. 7, Supplementary Data 3, Supplementary Discussion). All haplotypes in individuals from the Congolese lineage take basal positions in the class I clade, whereas the Upper Nile lineage haplotypes take basal positions in the class II clade.

To see if similar patterns of sorting of admixture variation could be detected elsewhere in the genomes, we tested if genomic SNPs that are highly differentiated among LVRS species were commonly derived from admixture between the two parental lineages. We inferred the origin of bi-allelic SNPs in six phenotypically diverse Lake Victoria species, the insectivore *Pundamilia pundamilia*, the insectivore/zooplankti-vore *P. nyererei*, the algae grazer *Neochromis omnicaeruleus*, the paedophage *Lipochromis melanopterus*, the insectivore *Paralabidochromis chilotes* and the piscivore *Harpagochromis cf. serranus* (Fig. 5). We found that at 30% of sites that are exceptionally strongly differentiated between Lake Victoria species (LV outliers), one of the alleles was indeed likely introduced into the ancestry of the radiation through Upper Nile lineage introgression (categories 3 + 4 in Fig. 5a, Supplementary Discussion). Sites at which the Congolese and Upper Nile lineage taxa are fixed for alternative alleles (category 4 in Fig. 5a) are enriched for LV outliers and show a pattern consistent with mosaic-like sorting of ancestral variants amongst Lake Victoria species (Fig. 5b). The enrichment for LV outliers in category 4 SNPs does not seem to be because of inherently increased fixation probability of these loci, as in pairwise comparisons of six control group cichlid species from

outside the radiation, LV outliers were not more often differentially fixed in the control groups than non-outliers among category 4 SNPs (Supplementary Table 5, Supplementary Discussion). The mosaic-like ancestry pattern is also corroborated by our whole-genome sequence data (Supplementary Fig. 8).

## Discussion

Here, we demonstrate that multiple large animal adaptive radiations arose from a hybrid swarm. Our results suggest ancient admixture between two lineages of haplochromines, one from the Upper Congo and one from the Upper Nile drainage, at the origin of the entire Lake Victoria Region Superflock of cichlid fish. Contemporary Upper Nile lineage introgression or earlier independent introgression events into each lake radiation are unlikely given (i) the highly supported genomic monophyly of the entire Lake Victoria Region Superflock (Fig. 1, Supplementary Fig. 1), (ii) similar Congo and Upper Nile lineage ancestry proportions in all lake radiations (Supplementary Fig. 6) and (iii) highly correlated Congolese and Upper Nile lineage ancestry tracts (Fig. 3). In addition, the fact that neither *H. gracilior* (Lake Kivu) nor *T. pharyngalis* (Lake Edward) show greater admixture into LVRS species that are sympatric with them than with allopatric members of the LVRS from other lakes (Fig. 2 and Supplementary Fig. 5) speaks against recent introgression. The radiations in Lakes Albert, Saka and Victoria have been geographically separated from Lakes Edward and Kivu, and hence from *T. pharyngalis* and *H. gracilior*, for between 4,000 (Saka) and 100,000 (Victoria) years[31,33,34] and show similar ancestry proportions.

Highly similar amounts of allele sharing between the LVRS species and both *H. gracilior* and *T. pharyngalis* (Supplementary Table 3, rows 2.1–2.4, Supplementary Fig. 3) suggests that the taxon that hybridized with the Congolese ancestor of the LVRS was a close relative of both Upper Nile species, or their ancestor. It is likely that the Congolese lineage colonized the Lake Victoria region through capture of Malagarasi (Congo) tributaries during a humid phase (for example, 145,000–120,000 years ago, Supplementary Fig. 3) and encountered representatives of the Upper Nile lineage that would by then have occupied the Lake Victoria region. The existence of large lakes in the Lake Victoria region at that time is likely to have provided ecological opportunity for diversification, which could be exploited by a genetically diverse hybrid swarm.

The finding that the two *LWS* opsin haplotype classes of the LVRS are each shared uniquely with one of the parental lineages (Fig. 4, Supplementary Fig. 7) suggests that the ancient Congo-Nilotic admixture event was the source of the high functional variation among LVRS cichlids at the *LWS* opsin gene[39,51]. Recombination between the ancestral haplotypes within the admixed populations further enhanced the functional variation of *LWS* haplotypes available for the LVRS radiation (Supplementary Discussion, Supplementary Data 3). This gene is particularly important for adaptation to the red-shifted end of ambient aquatic light spectra that characterize the waters of Lake Victoria[39,51]. Bringing together the Congolese and Upper Nile opsin haplotypes thus appears to have facilitated adaptation to an extreme range of light conditions and visual ecologies, with many transitions between them at both early as well as late stages of the radiation. Divergence into major ecologically differentiated clades (*Enterochromis* vs *Neochromis/Mbipia*), but also major habitat shifts between young sister species are associated with recruitment of Congolese versus Upper Nile lineage derived haplotypes at the *LWS* locus (Supplementary Discussion).

Our data suggest a much more general impact of the ancient admixture event on evolution in this set of young adaptive

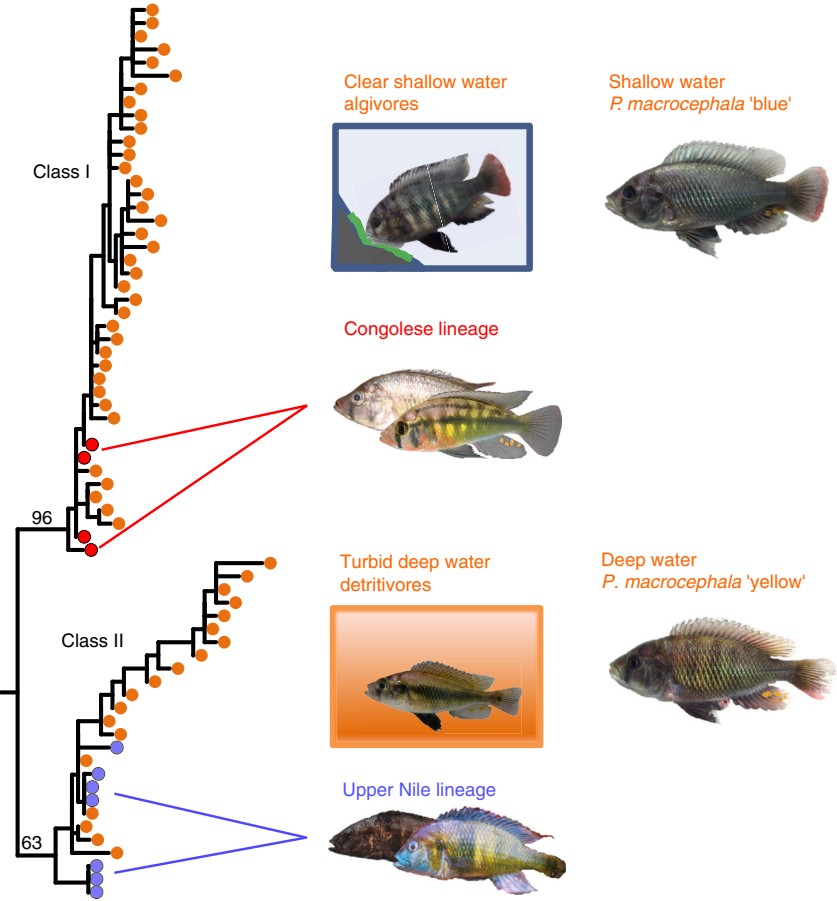

**Figure 4 | High *LWS* opsin diversity likely because of the ancient hybridization event.** The two major *LWS* opsin haplotype classes, I and II, in the LVRS (orange) are each shared exclusively with either the Congolese (*A. stappersi* and *A. sp.* 'Yaekama', red) or the Upper Nile lineage (*H. gracilior* and *T. pharyngalis*, blue), respectively (details in Supplementary Fig. 7; Supplementary Data 3). *LWS* haplotype class I is generally associated with cichlids living in shallow and clear water habitats, whereas class II is associated with deeper and more turbid habitats. Speciation by divergence in habitat type seems to have been accompanied by fixation of alternative *LWS* haplotypes both at early and at late stages of the adaptive radiation. This is exemplified by near fixation of alternative haplotype classes between ecologically divergent genera such as shallow water rocky shore algae scrapers of *Neochromis* and *Mbipia* versus mud bottom detritivores of the genus *Enterochromis*, and by the young incipient species pair of *Pundamilia macrocephala* 'blue' (living very shallow) and 'yellow' (living deeper) which have predominantly alleles of haplotype class I and II, respectively (Photo credits: Ole Seehausen, Adrian Indermaur, 'Teleos', Oliver Selz, Uli Schliewen).

radiations. Experimental crosses of cichlids have shown that intrinsic incompatibilities[38,56] and phenotypic novelty[57] both increase with genetic distance between the crossed species. Both kinds of variation could be important in speciation. The divergence time of the Congolese and Upper Nile lineage taxa, interpreted in the context of this previous experimental work, lets us predict that a hybrid swarm between these lineages would contain both intrinsic incompatibilities and transgressive trait variation. In agreement with this prediction, we found that genomic sites with alternative alleles fixed in the two parental lineages are enriched for outlier loci likely involved in divergent adaptation and species differentiation in Lake Victoria (Fig. 5). This suggests that speciation has commonly been associated with sorting of alleles brought together in the radiation ancestor by the admixture event. Sites showing negative epistatic interactions (Bateson Dobzhansky Muller (BDM) incompatibilities) would be expected to be fixed for alternative alleles in the parental lineages and thus fall in ancestry category 4 in Figure 5a. The strong enrichment of LV outliers in this category is in line with differential sorting of BDM incompatibilities among species that emerged from the hybrid swarm[21–23], which may have facilitated reproductive isolation among the members of the adaptive radiation. Given that the

correlation of Congo-Nilotic ancestry block patterns decreases between species with increasing phylogenetic distance within the LVRS (Fig. 3b), it is likely that Congolese and Upper Nile lineage alleles have been segregating in the LVRS for a long time and that they were progressively sorted among the radiation members during species diversification.

The combined evidence from the *LWS* opsin gene and the genome-wide patterns of sorting of ancestral variation is consistent with an important role of the ancestral hybridization for the evolution of the Lake Victoria Region Superflock. The resulting large genetic variation may help to explain how multiple adaptive radiations in the Lake Victoria Region, together forming 700 species, could arise in just 100–200 thousand years[31,32], including 500 + genetically and phenotypically well-differentiated species endemic to Lake Victoria which evolved in probably just 15,000 years[32,35]. Hence, our data provide evidence for the hypothesis[3] that hybridization between divergent lineages, when coincident with novel ecological opportunity such as colonisation of newly formed lakes, may facilitate rapid adaptive radiation through recombination and sorting of admixture-derived polymorphisms by natural and sexual selection.

That hybridization may fuel entire adaptive radiations has been hypothesized for several systems, but robust evidence has so far

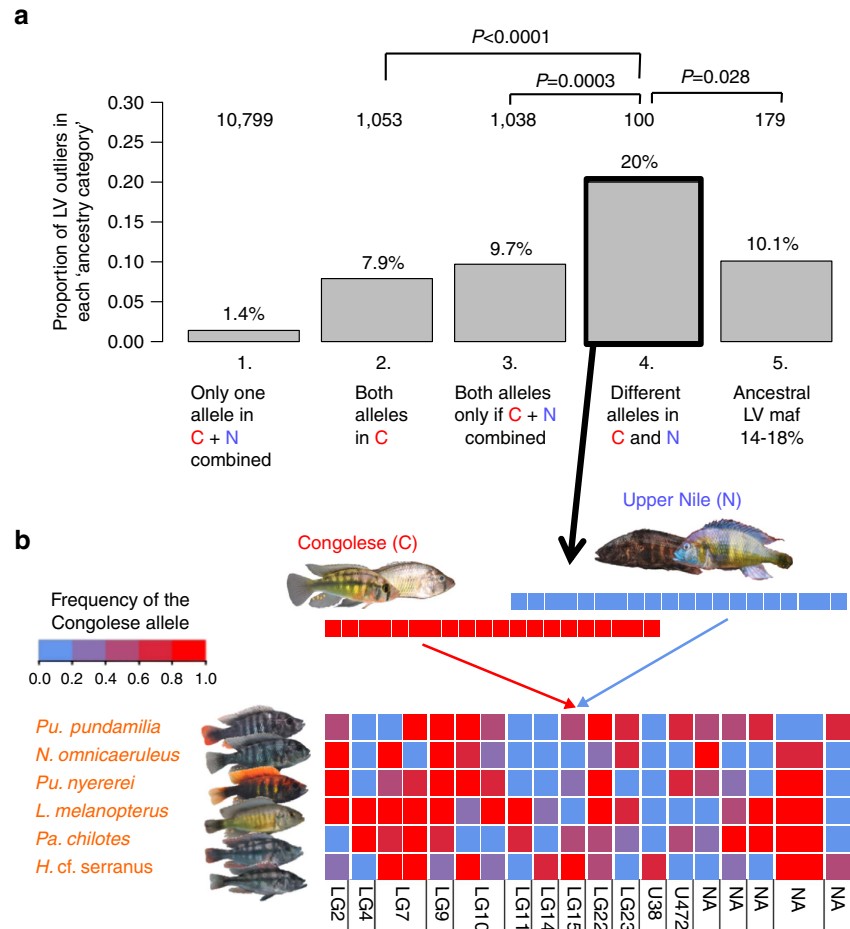

**Figure 5 | Differential sorting of hybridization-derived variation in the LVRS. (a)** Sites fixed for alternative alleles in the Congolese (C) and Upper Nile (N) taxa are enriched for high global $F_{ST}$ outlier SNPs in Lake Victoria. Of the 12,890 biallelic SNPs among six sympatric Lake Victoria species (shown in (**b**)), 340 are outliers of high global $F_{ST}$ (LV outliers). We assigned SNPs to five different ancestry categories according to the presence or absence of the two alleles in the Congolese (C) and Upper Nile (N) lineage taxa. The grey bars show the proportion of LV outliers among all SNPs in each ancestry category. Total SNP counts in each category and P-values of two-sided Fisher's exact tests are shown on top. Ancestry category (1) includes all SNPs for which only one of the two LV alleles was found in the Congolese and Upper Nile taxa together (novel LV allele or unsampled in parental lineages), (2) both LV alleles found in the Congolese taxa (polymorphic in LVRS even without Upper Nile hybridization), (3) only one allele found in Congolese but both alleles found in Upper Nile taxa (not available in LVRS without hybridization), and 4) Congolese and Upper Nile taxa each fixed for alternative LV alleles (not available in LVRS without hybridization) potentially including Bateson–Dobzhansky–Muller incompatibilities. Category 5 includes sites with similar initial allele frequency in Lake Victoria (16%) than sites fixed for alternative alleles in the parental lineages (category 4) to test if the enrichment in category 4 could simply be because of high initial allele frequency. (**b**) Differential sorting of parental alleles between Lake Victoria cichlid species at LV outliers fixed for alternative alleles in the Congolese and Upper Nile lineage taxa (mean global $F_{ST}$ among LV species = 0.52). Each square represents a SNP coloured according to the allele frequency in that species ranging from red (fixed for Congolese allele) to blue (fixed for Upper Nile allele). All except two sites (2 + 3 from the right) are located on different scaffolds of the *Pundamilia nyererei* reference genome. If known, chromosomal positions on the *Oreochromis niloticus* genome are shown below. At least 10 of the 22 chromosomes are involved in mosaic-like allele sorting between radiation species at loci that were fixed for alternative alleles in the parental lineages of the ancestral hybrid swarm (Photo credits: Oliver Selz, Ole Seehausen, Adrian Indermaur, 'Teleos', Uli Schliewen).

been confined to the allopolyploid Hawaiian silverswords[24]. Here, we report strong evidence that hybrid ancestry fuelled diversification in several large animal adaptive radiations. Thanks to the advent of high-throughput sequencing, our power to test for ancient hybridization and to study its impact on subsequent evolution has increased enormously. Future studies will reveal if hybridization in the ancestry of major adaptive radiations is widespread, and whether its occurrence may explain some of the observed large variation in the rates and volume of species diversification among lineages.

## Methods

**Experimental design.** To identify the closest relatives of the Lake Victoria Region Superflock (LVRS) and identify potential hybridization events, we performed restriction associated DNA (RAD) sequencing with haplochromine cichlid species from all major drainage systems that are either currently connected to the Lake Victoria region, or have been in the past, and representatives of all major radiations of the Lake Victoria Region Superflock. The closest relatives were inferred with maximum likelihood trees based on these data and on mitochondrial sequences of the same fish. D statistics, five population tests, and F4-ratio tests, were used to identify hybridization between lineages, determine the direction of gene flow and quantify ancestry proportions, respectively. Whole-genome sequencing was then performed with fish of the lineages identified as ancestral to the LVRS plus representatives of the superflock in order to corroborate the signatures of hybridization and to estimate ancestry block sizes.

**Taxonomic sampling.** We sampled haplochromines from all major lakes and several small lakes in the Lake Victoria region and from the Mpanga River that drains the Rwenzori Mountains in the drainage divide between Lakes Edward and Victoria (Supplementary Data 1, Fig. 1). Further, we collected or obtained

from other collectors haplochromines belonging to all phylogenetic lineages and from almost all African river drainages hosting haplochromines. We also included members of the haplochromine species flocks of the other African Great Lakes Malawi, Tanganyika and Mweru (Fig. 1, Supplementary Data 1). Samples were collected under research and sample export permissions of the Tanzania Fisheries Research Institute and the Tanzanian Ministry of Agriculture, Livestock and Fisheries; the Ugandan National Fisheries Resources Research Institute and Ministry of Agriculture, Animal Industry and Fisheries Uganda; and the Department of Fisheries Zambia. All samples were collected in compliance with applicable international and national guidelines for the use of animals, and ethical standards.

**RAD sequencing.** DNA was extracted from fin clips or muscle tissue with a standard phenol-chloroform-isoamyl alcohol extraction method[58]. Restriction-site Associated DNA sequencing (RADseq) was performed following a standard protocol[59]. Restriction digestion was done overnight (8–10 h) using the restriction endonuclease HF-*Sbf*I (NewEngland Biolabs) and 400–1,000 ng DNA per sample. P1 adaptors contained 5–8 bp long barcodes differing by at least two nucleotides from all other barcodes. The DNA was sheared with a Covaris S220 Focused-Ultra sonicator and fragments of 300–600 bp length were extracted from an agarose gel. We performed 18 PCR cycles to amplify the RAD fragments (30 s 98 °C, × 18 (10 s 98 °C, 30 s 65 °C, 30 s 72 °C), 5 min 72 °C). All libraries were single-end sequenced on an Illumina HiSeq 2,500 sequencer. The reads were de-multiplexed and trimmed to 84 nucleotides (nt, after barcode removal) with the process_radtags script from the Stacks pipeline[60], correcting single errors in the barcode and discarding reads with incomplete restriction sites. The length of 84 nt results from removing the barcode (mostly 6 nt long) and trimming off the last 10 nt because of reduced quality at the read ends. The FastX toolkit (http://hannonlab.cshl.edu/fastx_toolkit) was used to remove all reads containing at least one base with a Phred quality score below 10 and reads with more than 10% of bases with quality less than 30.

The reads of each individual were then mapped to the *Pundamilia nyererei* reference genome using Bowtie2 (ref. 61) with the end-to-end alignment option. Single nucleotide polymorphisms (SNPs) and genotypes were called using GATK Unified Genotyper v. 3.1 (ref. 62). All sites were then filtered with a custom-made Python script and vcftools v. 4.1 (ref. 63). SNPs within 5 nt from indels (insertions and deletions) were removed to avoid false SNPs because of misalignment problems, and SNPs were required to have a quality value of at least 30.

**Mitochondrial sequencing.** Two mitochondrial markers (*NADH* Dehydrogenase Subunit 2 (ND2)[64] using the primers ND2Met-F 5′-CAT ACC CCA AAC ATG TTG GT-3′ and ND2Trp-R 5′-GTS GST TTT CAC TCC CGC TTA-3′ and the mtDNA control region (D-loop)[65] with the primers FISHL15926-F 5′-GAG CGC CGG TCT TGT AA-3′ and FISH12s-R 5′-TGC GGA GAC TTG CAT GTG TAA G-3′ were amplified with PCR and Sanger sequenced for the same individuals or downloaded from GenBank for the same species as those included in the RADseq dataset (Supplementary Data 1). The sequences were aligned in ClustalW implemented in BioEdit 7.2.5 (ref. 66) and manually curated for correct local alignment.

**Phylogenetic analyses.** Phylogenies were reconstructed for the concatenated mitochondrial genes and the concatenated RAD sequences separately, including both variant and invariant sites using a maximum likelihood approach (RAxML v. 7.7.7 and ExaML v. 1.0.4)[67,68]. For the mitochondrial dataset, a maximum likelihood tree was reconstructed using three partitions, one for Dloop, one for the first and the second codon positions of ND2, and one for the third codon position of ND2. For each dataset, we performed a RAxML analysis with 100 rapid bootstraps using the GTRGAMMA model of rate heterogeneity. For the RAD-seq dataset, we used all concatenated sites with no more than 40 individuals missing (25%) to reconstruct a maximum likelihood tree with RAxML (ref. 67) and ExaML (ref. 68). Each of 100 bootstraps was performed by randomly sampling with replacement sites from the concatenated dataset to get a dataset of the original size. The maximum likelihood tree was then inferred for each resampled dataset with ExaML (ref. 68) using a GTRGAMMA model of rate heterogeneity, as recommended in the RAxML-light manual[68]. We calculated bootstrap support values based on these 100 topologies with RAxML (ref. 67). The nuclear and mitochondrial trees were rooted with the reference genome of *Oreochromis niloticus* and ladderized and plotted using the R-package Ape v. 3.1 (ref. 69).

We then used RAD-derived SNP data to infer the species tree for the LVRS groups and their closest relatives with SNAPP (ref. 45). SNAPP bypasses gene trees and computes species trees directly from independently inherited markers by integrating over all possible gene trees[45]. We restricted this analysis to two individuals per species (one for *A.* sp. 'Yaekama') and as SNAPP assumes no linkage among loci, we included only biallelic sites that were at least 500 kb apart from each other. The resulting data set contained 31 individuals and 1,817 sites. We ran SNAPP for 1,000,000 iterations, sampling every 1,000th iteration using default priors. We discarded the first 50% of the trees as burn-in and visualized

the posterior distribution of the remaining 500 trees as consensus trees in Densitree[70].

**Mitochondrial chronograms.** Dated phylogenies were reconstructed based on mitochondrial D-loop[65] and ND2 (ref. 64) sequences using BEAST v. 2.3.0 (ref. 71) and four different sets of calibration nodes (Supplementary Methods). We caution that the mitochondrial tree only shows the phylogeny of the maternal line and that time estimates more recent than one million years are most likely overestimates because of the increase of molecular rates towards the recent (Supplementary Methods).

To compare the splitting time between cichlid lineages with the major relevant geological events, we reconstructed paleogeographic maps at different time points based on previously published data and reviews (see Supplementary Methods).

**Patterson's D statistics.** To test for evidence of ancient admixture among lineages, we computed Patterson's D statistic[46,72] (ABBA-BABA test), a method to detect admixture based on the frequencies of discordant SNP genealogies in a four-taxon tree, with the software package ADMIXTOOLS v. 1.1 (ref. 48). Genotypes were discarded if they had less than 6 reads or a genotype quality Phred score <20 (that is, error probability >1%). Significance of D statistics was assessed with a block jackknife procedure using a z score of three standard errors as a threshold[48]. We used three individuals of *Astatotilapia flavijosephi* from Lake Kinneret as the outgroup population and we tested for evidence of gene flow from each Eastern and Upper Nile clade into the LVRS relative to allele sharing with the closest relative of the LVRS, the Congo drainage taxa *A. stappersi* from Zambia (Fig. 2) or *A.* sp. 'Yaekama' from the central Congo. For these analyses we used all individuals with at least 50% of the sites sequenced at a depth of at least 10 reads. To exclude the possibility that the D statistics are biased by the alignment of the reads to the Lake Victoria species *Pundamilia nyererei*, we also aligned the reads to the *Astatotilapia burtoni* reference genome, which is an outgroup to all taxa used in the D statistics. To rule out that the choice of outgroup (*A. flavijosephi*) biases the D statistics, we repeated the tests with *A. burtoni* or *A. desfontainii* as outgroups. All species combinations with number of individuals and SNPs included are given in Supplementary Table 2.

**Five population tests.** To infer directionality of gene flow between LVRS and the Nilotic taxa, we used an extended version of the partitioned D statistic test developed by Eaton & Ree[47] (we call it a 'five population test'). Similar to the D statistics, this test is based on discordant allele sharing patterns, but by considering five populations it allows one to infer the directionality of gene flow. The five taxa with the topology ((P1,P2), (P3a,P3b)),O include the potential source of gene flow (for example, P3a), the taxon receiving gene flow (for example, P1), a close relative of each of these two taxa (for example, P3b and P2) and an outgroup (O, see also Supplementary Table 3 for visualization). If genes had introgressed from for example, P3a into P1, a close relative (P3b) of the introgression donor (P3a) would also show excess allele sharing with P1, but to a lesser extent than P3a. This is because many derived alleles that introgressed from P3a into P1 will be shared by P3a and P3b because of their recent common ancestry. In contrast, we would not expect that a close relative (P2) of the receiver of gene flow, P1, would show excess allele sharing with P3a and P3b. In the genomic data, this would be seen as an excess number of BABBA patterns, where P1 shares a derived allele ('B') with both P3a and P3b, whereas P2 has the outgroup allele ('A'), as compared with the number of ABBBA patterns (P2 shares a derived allele with both P3 taxa). On the other hand, if the direction of gene flow was from P1 into P3a, P3b would not show excess allele sharing with P1 but instead, P2 would show excess allele sharing with P3a because of ancestrally shared alleles between P1 and P2 that introgressed into P3a. Therefore, gene flow from P1 into P3a would not affect the relative frequencies of BABBA and ABBBA patterns, but the pattern BBBAA (P1, P2 and P3a share a derived allele) would be more frequent than BBABA (P1 and P2 share a derived allele with P3b, see Supplementary Table 3). Counting discordant allele sharing patterns thus allows us to infer the direction of gene flow.

We computed the five population tests with a custom made script using the three individuals with least missing data for each LVRS group and the single individual with the most complete data for all other taxa. In contrast to the D statistics computed with ADMIXTOOLS, which are based on allele frequencies, our five population tests are calculated from a single individual for each focal population, following Eaton and Ree[47]. We tested each of the three individuals of each LVRS group separately, and report the means for each radiation. At heterozygous sites, one allele was chosen at random. For each combination of individuals tested, we counted the eight patterns needed to compute the four D statistics (Supplementary Table 3) using all sites without missing data. We calculated z scores in units of s.d. from 100 bootstrapped datasets (sites resampled with replacement) as in Eaton and Ree[47].

**Estimation of ancestry proportions.** The absolute value of D statistics does not only depend on the admixture proportion but also on the demographic history, the divergence between the hybrid parental lineages, and the genetic distance between

the real source taxon of gene flow and the sampled surrogate population[73]. Thus, the D statistic cannot be used directly to infer the magnitude of introgression. Instead, we applied the F4-ratio test[48,73] to infer the Upper Nile ancestry proportions in the different LVRS taxa using all genotypes with at least ten reads and SNPs with a maximum of 10% missing data (72,443 SNPs) with ADMIXTOOLS v. 1.1 (ref. 48). The F4-ratio test is based on four populations with the genealogy (((A,B)C)O) and a fifth potential hybrid population X that is tested for ancestry proportions from B and C (Supplementary Fig. 6). We repeated the test with different Eastern taxa used as population 'A', *T. pharyngalis* or *H. gracilior* as 'B', *A.* sp. 'Yaekama' as 'C' and *A. flavijosephi* as 'O' as shown in Supplementary Table 4. The Upper Nile ancestry proportions estimated with the combination of populations with the lowest s.d. were used for further analyses and to explain the F4-ratio test in Supplementary Fig. 6.

**Long-wave sensitive (LWS) opsin gene.** We sequenced the *LWS* opsin gene from exon 2 to 6 using the Carleton & Kocher primer combinations F2 (5′-TTT GAG GGT CCC AAT TAC CA-3′), R2 (5′-TCC ACA CAG CAA GGT AGC AC-3′) and F3 (5′-ACT GGC CTC ATG GAC TGA AG-3′) and R4 (5′-TCC CAA AAT GGA GAA CAT GG-3′) (http://cichlid.umd.edu/cichlidlabs/protocols/Basic/pcrprimer.html)[74]. The sequences were aligned together with all available Lake Victoria cichlid *LWS* sequences from GenBank using ClustalW implemented in BioEdit 7.2.5 (ref. 66) and then manually curated. For sequences used, see Supplementary Data 3. A maximum likelihood tree was reconstructed including both variant and invariant sites using the GTRGAMMA model of rate heterogeneity with RAxML v. 7.7.7 (ref. 67). We used one partition with concatenated introns, one for the first plus second codon position of concatenated exons, and one for the third codon positions. The tree was rooted with the *Oreochromis niloticus* sequence and nodal support values were drawn from 100 rapid bootstraps performed with RAxML. The tree was ladderized and plotted using the R-package Ape v. 3.1 (ref. 69).

To visually assess the effects of recombination on the *LWS* opsin alleles in the Lake Victoria Region cichlids, we extracted all positions of the alignment that were polymorphic among the LVRS sequences and also differed between the Congolese and Upper Nile taxa. If one of the alleles was only found in the Congolese taxa but not in the Upper Nile taxa, we coloured it red, whereas alleles found only in the Upper Nile taxa were coloured blue. Many LVRS haplotypes could be explained by recombination between the two divergent allele classes occurring in the Congo and Upper Nile clades. The second *LWS* gene tree was reconstructed with all recombinant sequences falling between the two major allele classes in the phylogenetic tree removed from the alignment.

For information about the occurrence of *LWS* haplotypes across LVRS species, we compiled data from Tables 1 + 2 in Terai *et al.*[51], Fig. 1 and Supplementary Table 4 in Terai *et al.*[49], Supplementary Fig. 4b in Seehausen *et al.*[39], Figure S3 in Miyagi *et al.*[54] and additional GenBank sequences by Carleton *et al.* (GenBank). Information sources and GenBank accession numbers are provided in Supplementary Data 3.

**Sorting of ancestral alleles.** We created a RAD sequence dataset of our Congolese and Upper Nile LVRS relatives and previously published sequences[36] of six sympatrically occurring Lake Victoria species that are ecologically and phenotypically diverse (*Pundamilia pundamilia* (n = 10, blue male nuptial coloration, benthic insectivore-omnivore), *P. nyererei* (n = 11, red planktivore-omnivore), *Harpagochromis* cf. *serranus* (n = 10, blue piscivore), *Lipochromis melanopterus* (n = 9, yellow paedophage), *Paralabidochromis chilotes* (n = 9, blue benthic insectivore) and *Neochromis omnicaeruleus* (n = 12, blue algae scraper)). The sites were filtered for having at least 5 individuals covered in each species at a sequencing depth of at least six reads. We used Arlequin v. 3.514 (ref. 75) to calculate global $F_{ST}$ values based on a hierarchical island model among the six Lake Victoria cichlid species. We calculated allele frequencies for the Congolese (*Astatotilapia stappersi* from Zambia and *A.* sp. 'Yaekama' from DRC, n = 5) and the Upper Nile taxa ('*Haplochromis*' *gracilior* and *Thoracochromis pharyngalis*, n = 9) to get estimates of the allele frequencies for the parental lineages of the LVRS and to infer the likely ancestry of the alleles segregating in Lake Victoria. We excluded sites monomorphic in the subset of Lake Victoria species or sequenced in less than three individuals in either the Congolese or the Upper Nile lineage taxa. We extracted sites for which only one of the two alleles segregating in Lake Victoria cichlids were found in the Congolese and Upper Nile lineage taxa together (category 1). Next, we extracted sites with both alleles segregating in the Congolese taxa representing ancestral standing genetic variation that would have been present in the Lake Victoria radiation also without ancient hybridization with the Upper Nile lineage (category 2).

We identified Lake Victoria polymorphic sites for which only one allele was found in the Congolese taxa, and the second allele was found exclusively in the Upper Nile taxa, representing genetic variation that can be attributed to the ancient admixture event (category 3). We also identified the subset of sites fixed for alternative alleles in the Congolese and Upper Nile taxa (category 4). These are sites that could be involved in epistatic BDM-like incompatibilities between Congolese and Upper Nile alleles. Sites fixed for alternative alleles in the Congolese and Upper Nile lineage taxa are expected to have been present at the onset of the Lake Victoria radiation at a mean minor allele frequency of 16% (that is, the

inferred Upper Nile ancestry proportion for the Lake Victoria radiation). To take the effect of this moderately high initial allele frequency into account in our subsequent analyses, we created a dataset of SNPs with comparable expected ancestral allele frequency in Lake Victoria (category 5) by weighting the allele frequencies of the Congolese and Upper Nile lineage taxa at each site by 0.84 and 0.16, respectively, corresponding to the inferred ancestry proportions for the Lake Victoria radiation. Of these, we extracted all sites with a weighted minor allele frequency of 14–18% excluding sites that are divergently fixed between the Congolese and Upper Nile taxa. As an example, if at a given bi-allelic site the frequency of one allele was 0.1 in the Congolese taxa and the frequency of the same allele was 0.5 in the Upper Nile taxa, we calculated $0.1 \times 0.84 + 0.5 \times 0.16 = 0.164$. As this site would have a weighted minor allele frequency between 14% and 18%, it would be included in the control set of SNPs. Note that such loci are not predicted to be involved in BDM incompatibilities because all their alleles were present within at least one of the two parental populations. The proportion of sites that were outliers of high global $F_{ST}$ (P < 0.05) among the Lake Victoria species ('LV outliers') was calculated for each dataset and compared between datasets using a two-sided Fisher's exact test. The robustness of the test to erroneously inferred ancestry proportions was assessed by repeating the calculations with different Upper Nile allele frequency weighting, ranging from 10 to 30% (Supplementary Discussion).

To check if LV outliers fixed for alternative alleles in the parental lineages inherently have a higher fixation probability than non-outliers, we analysed fixation patterns in other closely related cichlid species that are not part of the radiation. We used *Astatotilapia bloyeti*, *A. sparsidens*, *A. flavijosephi*, *A. calliptera*, *A. burtoni* and *Astatoreochromis alluaudi* as we have at least three sequenced individuals of each. For each pairwise comparison between these species, we extracted sites with at least three genotypes sequenced at a minimum depth of six reads in both species. We then checked if sites fixed for alternative alleles in the Congolese and Upper Nile lineage taxa were also fixed for alternative alleles between these other species. Finally, we tested with Fisher's exact tests if sites that were high global $F_{ST}$ outliers among Lake Victoria species were also more often differentially fixed between these other species than non-outlier sites (see also Supplementary Discussion).

To analyse the genomic distribution of LV outliers that are fixed for alternative alleles between the Congolese and Upper Nile taxa, we used liftOver (http://genome.ucsc.edu/) with a chain file from Brawand *et al.*[50] to get positions on the *Oreochromis niloticus* genome which includes chromosomal information.

**Whole-genome sequencing data.** To corroborate our findings from RAD sequence data, and for the analysis of ancestry block sizes, we sequenced whole genomes of nine fish from Lake Victoria, two from Lake Kivu, and of the taxa that we found to be the closest extant representatives of the lineages directly ancestral to the LVRS, the Congolese *Astatotilapia stapperi*, and the Upper Nile *Thoracochromis pharyngalis* and '*Haplochromis*' *gracilior*. Whole-genome sequencing data was generated using PCR-free library preparation[76] and Illumina HiSeq 3000 paired-end sequencing for 11 individuals. Three additional individuals, sequenced the same way on the same machine, were taken from McGee *et al.*[77] (see Supplementary Data 1 for sample information). Local alignment against the *Astatotilapia burtoni* reference genome[50] was performed with Bowtie 2 (ref. 61). For variant calling and genotyping we used Haplotype Caller (GATK v. 3.5)[62]. Genotypes with fewer than five reads, multiallelic sites, and indels were removed using vcftools v. 4.1 (ref. 63).

Whole-genome D statistics were calculated with ADMIXTOOLS v. 1.1 (ref. 48) using all biallelic sites with a 1% minor allele frequency cutoff and a maximum missing data proportion of 20% across all 14 genomes. The reference genome (*A. burtoni*) was used as an outgroup.

To study signatures of admixture along the genome, we used $f_d$ statistics by Martin *et al.*[78]. In comparison to D statistics, $f_d$ is more suited for small genomic regions[78]. ABBA and BABA pattern counts were calculated using allele frequencies by weighting each segregating site according to its fit to the ABBA or BABA pattern[46,72,78]. As an example, if a site has derived allele frequencies of 0, 0.5, 1 and 0 in P1, P2, P3 and the outgroup, respectively, it would count as half ABBA site. We used *A. stappersi* as P1, different LVRS individuals as P2, *H. gracilior* and *T. pharyngalis* as P3 and *A. burtoni* as outgroup (Supplementary Fig. 8). Note that P1 and P2 are switched as compared to D statistics, where LVRS are used as P1 and the Congolese lineage as P2. This difference is simply for consistency with the description of $f_d$ in Martin *et al.*[78], and thus here ABBA represents sites with a derived allele shared between LVRS and Upper Nile, whereas BABA represents sites with a derived allele shared between Congolese and Upper Nile lineage taxa. $f_d$ is calculated as the difference between ABBA and BABA patterns compared to the maximum possible difference where gene flow between P2 and P3 would equal random mating[78]. Positive $f_d$ values indicate gene flow between P2 (LVRS) and P3 (Upper Nile). We calculated $f_d$ in non-overlapping sliding windows of 10 kb along the *A. burtoni* scaffolds using the Python script by Martin *et al.*[78]. Windows with less than five total ABBA and BABA patterns were excluded. As only positive $f_d$ values are indicative of excess allele sharing between Upper Nile and LVRS, and are correctly standardized, values with negative D scores were set to 0 as in Martin *et al.*[78]. Pearson's product-moment correlations of $f_d$ values between different individuals were calculated with the stats R-package.

Putative Congolese or Upper Nile ancestry was assessed by comparing the frequency of ABBA sites (LVRS shares the derived allele exclusively with Upper Nile taxa) with the frequency of BBAA sites (LVRS shares the derived allele exclusively with the Congolese lineage representative *A. stappersi*) in non-overlapping sliding windows of 3 kb. Scaffolds with <100 kb data were removed. As for the $f_d$ statistic, ABBA and BBAA pattern counts were calculated using allele frequencies by weighting each segregating site according to its fit to the ABBA or BBAA pattern. Only windows with a total ABBA and BBAA pattern count exceeding one were used. Windows with a minimum ABBA proportion (ABBA/(ABBA + BBAA)) of 0.7 were defined as candidate windows of Upper Nile lineage ancestry and are coloured in blue, whereas genomic regions with an ABBA proportion of 0.3 or less were defined as putative Congolese lineage derived windows and are highlighted in red. Ancestry tracts were defined as consecutive sliding windows of the same colour (red or blue) ignoring single sliding windows without data. Expected ancestry block sizes were calculated using a formula from Racimo et al.[79]. Assuming ∼50,000–100,000 generations since admixture, a recombination rate of $2.5 \times 10^{-8}$ and an Upper Nile proportion of 20%, the expected mean length of the admixture tracts is $(0.8 \times 2.5 \times 10^{-8} \times (50,000 \text{ to } 100,000) - 1))^{-1} = 500$ bp to 1 kb (ref. 79). The breakup of ancestry blocks may have slowed as the original hybrid population began to undergo genomic stabilization, speciated and formed geographically isolated radiations in separate lakes each of which underwent further stabilization independent of each other, likely associated with differential sorting of the ancestral variation.

**Data availability.** Mitochondrial and *LWS* opsin sequences are available on GenBank under the accession numbers KY366716-KY366843 for ND2, KY366844-KY366970 for D-loop, and KY366971-KY366986 for *LWS* opsin sequences. RADseq and whole-genome sequencing reads generated in this study can be downloaded from the NCBI Sequence Read Archive under Bioproject PRJNA355227. The JAVA program for 5-population tests is publicly available on GitHub (https://github.com/joanam/scripts).

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

## Acknowledgements

We thank all collaborators who provided tissue samples over the years: Sigal Balshine-Earn (McMaster), Roger Bills (SAIAB), Julia Day (UCL), Yves Fermon (Paris), John Friel (CUMV), Martin Genner (U Bristol), Sylvain Piry (Paris), Lukas Rüber (Bern Natural History Museum), Walter Salzburger (U Basel), Uli Schliewen (Zoologische Staatssammlung Munich), Erwin Schraml (Augsburg), Jos Snoeks (Africa Museum Tervuren), Melanie Stiassny (AMNH), George Turner (U Bangor), Sylvester Wandera (NAFIRI; Uganda) and Marco Welss (Kressberg). We also thank Matt McGee for help with DNA extraction, Keith Harshman of the Lausanne Genomic Technologies Facility and Cord Drogemüller, Tosso Leeb, Muriel Fragnière and Michèle Ackermann of the NGS platform of the University of Bern for Illumina sequencing support, Aria Minder and Stefan Zoller of the Genetic Diversity Center (GDC) at ETH Zürich and Irene Keller of the Interfaculty Bioinformatics Unit at the University of Bern for lab and bioinformatics support and discussion, and Andy Cohen for comments on the paleogeographic maps. This research was supported by the Swiss National Science Foundation grant PDFMP3 134657 to O.S. and L.E.

## Author contributions

O.S., J.I.M. and C.E.W. designed the study; O.S. gathered and identified the cichlid samples; J.I.M., S.M. and D.A.M. performed the lab work (DNA extraction, mtDNA and *LWS* sequencing and RAD library preparation); J.I.M. conducted the analyses, with assistance from O.S., C.E.W., D.A.M. and L.E.; J.I.M. prepared the manuscript together with O.S. and C.E.W. and L.E. and D.A.M. contributed to writing.

## Additional information

**Competing financial interests:** The authors declare no competing financial interests.

