## [Peer Review File · Nature Communications]

Reviewers' comments:

Reviewer #1 (Remarks to the Author):

I have not reviewed the paper previously, but have gone through the manuscript and the authors' responses to previous comments. By and large it seems that the authors have done an excellent job at addressing the comments. The manuscript is very well written, and technically about as solid as it could be. The major advance here is that it provides a demonstration of a phenomenon that has long been thought to happen, though providing evidence has been difficult. The authors provide perhaps the best evidence yet, so this is a tremendously important and novel paper.

The main comment I have is that I think that the authors could do a lot better at setting up the broader context of the study. While elegant and very important, the current focus is very heavily on cichlids. It looks like this point has been noted in previous versions of the paper, and the authors now address this by extending their comparisons to other hybridization studies and have added a paragraph that provides a broader context. However, this doesn't come until the last paragraph of the Discussion.

The point here is that I think that the rather narrow focus of the paper undermines its importance. The role of admixture in early diversification is not new to studies of adaptive radiation. Carlquist (1974) argued that natural hybridization can be a constructive force in the evolution of the waif flora. In Hawaiian silverswords, it has been shown that all crosses yield vigorous hybrids and studies again have long suggested that this phenomenon may have been important in the adaptive radiation of many plants in Hawaii (Baldwin and Wagner 2010). Carson et al. (1990) argued famously for the importance of the dynamic landscape of Hawaii created by lava flows, with repeated cycles of isolation and admixture of populations (or species) in new combinations (as a result of lava flows and subsequent vegetation regeneration) potentially providing an "evolutionary crucible" that can facilitate and potentially accelerate diversification (Gillespie 2016). And, of course, the arguments raised for Hawaii have been echoed in the Galapagos, with Grant suggesting that species of Galapagos finches undergo a fusion/fission cycle.

Almost all of these studies have pointed to the need for further work to understand how genetic changes interact with a changing set of ecological interactions across a shifting mosaic of landscapes to promote species diversification in the context of adaptive radiation. The current study by Meier et al goes a long way towards filling that void of information.

Specific points

Line 78 "hybridization between relatively distantly related species, when coincident with

ecological opportunity" One of the major keys to this process is the "relatively". There appears to be some kind of specific time period below which there is little effect, and above which, taxa have formed new species. Krehenwinkel and Tautz recently showed that northern expansion of the wasp spider, *A. bruennichi*, into a novel climatic niche was the result of the coming together of different populations that had been isolated for an extended period. So this phenomenon has been documented in other systems.

This is also wrapped up in the idea of admixture (usually thought to occur between populations) and hybridization (usually thought to occur between species). Obviously, when comparing vertebrate radiations with those of plants and arthropods, this distinction becomes very obscure (many vertebrate "species" would likely be called populations in plants or arthropods Though it may also happen the other way round, but the point is that there is a lot of wiggle room).

Line 230 "Some of these sites that are fixed for alternative alleles in the parental lineages may be involved in negative epistatic interactions when brought together in the hybrid swarm" Note that this is a phenomenon that Carson (1990) alluded to.

Line 250 "However, finding evidence that hybridization between distantly related species may facilitate the onset of an adaptive radiation⁴ has been more challenging." Hugely challenging, though widely discussed. Would be great to highlight the early discussions more, and that they basically reached stalemate because they couldn't be tested

Another point here is the "distantly related". This relates to the point regarding the "relatively distantly related species" above, and also makes comparison between radiations of disparate taxa really difficult. The thing is, the species are only distantly related relative to the extremely closely related species swarm that comprises the radiation.

Line 247 "Hybridization among members of adaptive radiations has been shown to facilitate additional speciation events in many systems including Darwin's finches, *Heliconius* butterflies, *Rhagoletis* fruit flies, sunflowers, Lord Howe *Coprosoma* ...". These are very different systems, and only some would be considered adaptive radiations. Rather than simply list these examples, would be good to highlight exactly what elements of the phenomenon are shown in these different systems – what is the same and what different. Note that this does not require extended discussion of each system, but rather simply placing the different studies in the context of what they actually bring to the table for understanding the role of hybridization and admixture in adaptive radiation.

Line 254 "Hybridization between distantly related species prior to the beginning of adaptive radiation has also been demonstrated in a plant radiation, the Hawaiian silverswords " This is the first mention of the phenomenon being well documented in another radiation. But that was a long time ago, and there were major limitations in this study. The point here is that people have been looking for concrete evidence of the phenomenon for a very long time, and this study by Meier et al is perhaps the first to provide really solid data as to how it might happen, and under what circumstances.

Reviewer #2 (Remarks to the Author):

This paper uses genomic data to argue that the Lake Victoria region superflock of haplochromine cichlids initiated as a result of a hybridization event between a species or species from the upper Nile valley, and others from the Congo region. In general I found this study convincing, but in places I found the reasoning and evidence frustratingly difficult to understand. Part of this is due to the brevity of the exposition, coupled with the complexity of multi-species, whole-genome evidence. In my view an article like this should steer away from discussions that require the reader de-coding obscure forms of explanation in figures that are there only in order to fit a lot of complexity into the slim format of a Nature-style article. For instance it took me a long time to work out what b, s, p, w, t, g were in Fig. 2b; I think at least some of these letters and their explanations should be in the legend, and perhaps in the text as examples, where I hunted for it, only to find it eventually in tiny parentheses in Fig. 2a.

I'm unclear why the relatively weak mitochondrial information was deemed worth including, given that the vast weight of the phylogenetic evidence is from the genomic resequencing (RAD) data. Cutting this would save some space. The relatively short sequence lengths studied also would be prone to phylogenetic error. For similar reasons, the relatively weak evidence for time-dependent rates of molecular evolution seems a red herring here (lines 367-373), especially as one can't do anything about it in the dated tree. Much of the evidence may anyway be spurious due to difficulties with finding adequate priors for Bayesian phylogenetics, and the tendency for the tips of trees to be elongated in many programs. In other words it can be an artefact due to Bayesian mis-specification (see Brent Emerson 2007 *Syst Biol.* 56, 337–345).

l. 128: here I'd mention that these two species are g and t in Fig. 2b.

One problem with the analysis is that the RAD loci are largely anonymous, and so little information is gained as to the mechanism of speciation or the contributing genes to rapid radiation. To some extent a fuller discussion of the the LWS opsin results would ameliorate this, but these data is treated in frustratingly little detail, and all of the functional information seems to be referring to other papers rather than being spelt out clearly here. We are more or less told the LWS opsin an important locus in ecology and evolution, and in adaptation to depth and turbidity, but the mechanisms are unclear. The mention in the results is brief, and there is nothing about it in the discussion, and the supplementary discussion does not add much. And yet a whole figure is devoted to this in the main text. I was left with many questions. Why would a shift in wavelength sensitivity be important in adaptation to differential exploitation of different light environments? How might this locus affect mate choice and sexual selection (as opposed to being told there are citations to data that it does affect it)? Are ALL the Nile species known (not just the two mentioned) fixed for the class II allele, and all the Congo species known to be fixed for the class I allele? If so, how do these ecologically important alleles fit their environments in the presumed ancestors? In other words, why might there be a difference in the first place? Adaptation?

Historical accident? Why are the allelic classes referred to variously Class I vs. Class II (Figs. 4, S7 and supplementary discussion), H and L (Supplementary discussion), and Congolese and Upper Nile (supplementary discussion).

lines 230-233: "Some of these sites that are fixed for alternative alleles in the parental lineages may be involved in negative epistatic interactions when brought together in the hybrid swarm and differential sorting of such Bateson-Dobzhansky-Muller incompatibilities may facilitate reproductive isolation among the species emerging from the hybrid swarm." It's certainly possible but it is hardly a finding of this paper.

lines 263-265: " If hybridization prior to the onset of adaptive radiation is typical for particularly rapid and large species radiations, the importance of interspecific hybridization in the evolution of species diversity may have been underestimated". This is certainly true, but I was tempted to say, yes, but if it's not typical, then it may not have been underestimated! In other words, this is a rather empty statement with which to end a paper.

l. 299: Why truncate at 84 nucleotides? I take it that the entire RAD sequence was used, and not just a single SNP from each restriction site? Why not paired end? And did you use each site for both forward and reverse directions of read from each restriction site after aligning to the reference, yielding a ~150 bp total read for each rad locus?

lines 481 to 505. This method seems to cover only a few of the "categories" of ancestral origin of fig. 5 (i.e. categories 2-4). If possible, can some of these categories be lumped to make the whole easier to understand?

Fig. 5: I don't fully understand this figure. Delete Bateson-Dobzhansky-Muller incompatibilities, for which there is no evidence given, from the legend. From the description, I would have thought category 6 should be a bigger category than category 5, not smaller as shown. Overall I was confused by these categories and what they're supposed to show us; can they be reduced to a more manageable number. I would have thought that the "neutral" expectation that the allele frequency was approx. equal (4-18%) to the estimated ancestral fraction in the ancestor is prone to all sorts of other things like drift, and so I think that "polymorphic" would be a better overall category. Finally, are there really only 20 sites fixed for alternative alleles between the Congolese and Upper Nile species as in Fig. 5b?

Point-by-point response

Reviewer #1

Comment 1.1: *I have not reviewed the paper previously, but have gone through the manuscript and the authors' responses to previous comments. By and large it seems that the authors have done an excellent job at addressing the comments. The manuscript is very well written, and technically about as solid as it could be. The major advance here is that it provides a demonstration of a phenomenon that has long been thought to happen, though providing evidence has been difficult. The authors provide perhaps the best evidence yet, so this is a tremendously important and novel paper.*

>> We are pleased that the reviewer finds our study to be of high importance and novelty. We appreciate it a lot that he/she took the time to read through all the comments of the previous rounds of review, and we are happy that he/she is satisfied with our responses. We are also glad that the reviewer likes our writing and analyses.

Comment 1.2: *The main comment I have is that I think that the authors could do a lot better at setting up the broader context of the study. While elegant and very important, the current focus is very heavily on cichlids. It looks like this point has been noted in previous versions of the paper, and the authors now address this by extending their comparisons to other hybridization studies and have added a paragraph that provides a broader context. However, this doesn't come until the last paragraph of the Discussion.*

The point here is that I think that the rather narrow focus of the paper undermines its importance. The role of admixture in early diversification is not new to studies of adaptive radiation. Carlquist (1974) argued that natural hybridization can be a constructive force in the evolution of the waif flora. In Hawaiian silverswords, it has been shown that all crosses yield vigorous hybrids and studies again have long suggested that this phenomenon may have been important in the adaptive radiation of many plants in Hawaii (Baldwin and Wagner 2010). Carson et al. (1990) argued famously for the importance of the dynamic landscape of Hawaii created by lava flows, with repeated cycles of isolation and admixture of populations (or species) in new combinations (as a result of lava flows and subsequent vegetation regeneration) potentially providing an "evolutionary crucible" that can facilitate and potentially accelerate diversification (Gillespie 2016). And, of course, the arguments raised for Hawaii have been echoed in the Galapagos, with Grant suggesting that species of Galapagos finches undergo a fusion/fission cycle.

Almost all of these studies have pointed to the need for further work to understand how genetic changes interact with a changing set of ecological interactions across a shifting mosaic of landscapes to promote species diversification in the context of adaptive radiation. The current study by Meier et al goes a long way towards filling that void of information.

>> We thank the reviewer for this helpful perspective and for pointing us to additional literature on the possible impact of hybridization on adaptive radiation. We are pleased that the reviewer agrees that our study goes beyond previous studies in showing that hybridization facilitated adaptive radiation. We would like to point out though, that the Hawaiian and Galapagos fusion-fission cycles, for as far as the existing literature is concerned, are examples of hybridization among radiation members possibly facilitating further speciation events in an already ongoing adaptive radiation ("syngameon hypothesis" of Seehausen, 2004). Our study, on the other hand, provides evidence for hybridization between distant lineages fuelling the onset of an adaptive radiation ("hybrid origin of adaptive radiation" of Seehausen 2004). Apart from the Hawaiian silversword alliance, this has not been robustly demonstrated in nature to the best of our knowledge even though it has been hypothesized several times (e.g. for Lake Malawi cichlids, alpine whitefish, and

Hawaiian mints) and is thus likely of broad relevance. In the revised version of our manuscript, we have now extended the Introduction to give a more extensive overview of previous work on the role of hybridization in adaptive radiation, and we review the distinction of the “syngameon hypothesis” and the “hybrid swarm origin of adaptive radiation” hypothesis, which places our study in a broader context early in the paper, as suggested by the reviewer.

We now write in the Introduction (lines 50-73):

“Hybridization among members of an adaptive radiation has been shown to sometimes facilitate additional speciation events, boosting the growth of a radiation (“syngameon hypothesis” Seehausen 2004). Introgression of traits involved in divergent adaptation or reproductive isolation has been demonstrated in multiple iconic adaptive radiations (e.g. traits related to host shift in *Rhagoletis* fruit flies (Feder, Xie et al. 2005), wing patterns in *Heliconius* butterflies (Gilbert 2003; Pardo-Diaz, Salazar et al. 2012; The Heliconius Genome Consortium 2012; Wallbank, Baxter et al. 2016), or beak shape in Darwin’s finches (Lamichhaney, Berglund et al. 2015)). In other systems, the hybrid ancestry of some radiation members has been inferred, but a direct link between introgressive hybridization and relevant traits awaits further testing (e.g. sailfin silversides (Herder, Nolte et al. 2006), and cichlid fishes of Lakes Tanganyika (Salzburger, Baric et al. 2002; Weiss, Cotterill et al. 2015; Meyer, Matschiner et al. 2016), Malawi (Genner and Turner 2012), Victoria (Keller, Wagner et al. 2013; Meier, Sousa et al. 2016), and Barombi Mbo (Schliewen and Klee 2004).

Another hypothesis for a perhaps more fundamental role of hybridization in adaptive radiation, distinct from the syngameon hypothesis, is the idea that hybridization may seed the onset of an entire adaptive radiation (Seehausen 2004). This hypothesis has been more challenging to test. So far the only adaptive radiation for which a hybrid origin has been robustly demonstrated is the Hawaiian silverswords, which have radiated from an allopolyploid hybrid population between two North American tarweed species (Barrier, Baldwin et al. 1999). Because gene and genome duplication are also proposed to facilitate adaptive radiation (Lynch and Conery 2000), it is difficult though to distinguish between effects of hybridization per se and those of gene or genome duplication in this case. Evidence consistent with a hybrid swarm origin of entire radiations has also been found in Alpine whitefish (Hudson, Vonlanthen et al. 2011), the “mbuna” group of the Lake Malawi cichlid fish radiation (Joyce, Lunt et al. 2011), and allopolyploid Hawaiian endemic mints (Lindqvist and Albert 2002; Lindqvist, Motley et al. 2003) and possibly other polyploid plant radiations on Hawaii (Baldwin and Wagner 2010). However, it remains to be tested if in these systems hybridization occurred prior to the onset of the radiation or after the radiation had already unfolded, and if hybridization-derived polymorphisms were involved in speciation and adaptive diversification.”

We thank the reviewer for pointing us to the early literature suggesting that hybridization may be a “constructive force in evolution”. We now cite this work at the beginning of the second paragraph of the Introduction, at the first mentioning of hybridization: “Hybridization between species can instantaneously boost genetic variation which may facilitate speciation and adaptive radiation (Anderson and Stebbins 1954; Carlquist 1974; Carson, Lockwood et al. 1990; Rieseberg 1997; Seehausen 2004; Mallet 2007; Abbott, Albach et al. 2013; Arnold 2015)” (lines 38-39).

Specific points

Comment 1.3: *Line 78 “hybridization between relatively distantly related species, when coincident with ecological opportunity” One of the major keys to this process is the “relatively”. There appears to be some kind of specific time period below which there is little effect, and above which, taxa have formed new species. Krehenwinkel and Tautz recently showed that northern expansion of the wasp spider, A. bruennichi, into a novel climatic niche was the result of the coming together of different populations that had been isolated for an extended period. So this phenomenon has been*

documented in other systems.

>> The reviewer raises two important points connected to our study. First, we agree that ecological opportunity and hybridization must coincide in order for the hybridization to impact the diversification trajectory of the lineage. In the case of our study, we believe that this interaction was realized through the existence of large lakes in the Lake Victoria region around the time as the Nile and Congolese lineages came together in this region (see Figure S3). In the Discussion, after “It is likely that the Congolese lineage colonized the Lake Victoria region through capture of Malagarasi (Congo) tributaries during a humid phase (e.g. 145,000-120,000 years ago, Supplementary Fig. 3) and encountered representatives of the Upper Nile lineage that would have hitherto occupied the Lake Victoria region.”, we now write “The existence of large lakes in the Lake Victoria region at that time likely provided ecological opportunity for diversification that could be exploited by a genetically diverse hybrid swarm” (lines 255-257).

Second, if the taxa which hybridize upon secondary contact are too closely related, the hybridisation event may not generate much novel diversity, but if they are too distantly related, hybrid fitness will be so low that the hybrid population is unlikely to establish. Previous experimental work in our group was designed to test these ideas and we have identified the location and width of this “window of opportunity” for cichlids. Stelkens and Seehausen (2009) showed that the probability of generating novel traits in hybrids between cichlid lineages indeed increases with divergence time. Stelkens, Young et al. (2010) and Stelkens, Schmid et al. (2015), on the other hand, showed that this window of opportunity begins to close after about 5 million years of divergence because the hybrid fertility and viability reach very low levels. As long as incompatibilities between the parental taxa do not completely prevent the formation of fertile hybrids, they may promote the evolution of reproductive isolation among new species emerging from a hybrid swarm through differential sorting of these incompatibilities in the species emerging from the hybrid swarm (Seehausen 2013; Hermansen, Haas et al. 2014; Schumer, Cui et al. 2015). We thus expect that intermediate divergence between the parental lineages most readily facilitates the onset of adaptive radiation. The divergence that we report here between the ancestors of the hybrid population that seeded the LVRS radiation, falls right within the window of opportunity identified experimentally by Stelkens and Seehausen (2009), Selz, Thommen et al. (2014), Stelkens, Young et al. (2010) and Stelkens, Schmid et al. (2015). This supports the idea that these lineages were sufficiently distantly related to produce hybrids with large trait variation and some incompatibilities but not too many. We now write in the Discussion (lines 272-278): “However, our data suggest a much more general impact of the ancient admixture event on evolution in this young adaptive radiation. Experimental crosses of cichlids have shown that intrinsic incompatibilities (Stelkens, Young et al. 2010; Stelkens, Schmid et al. 2015) and phenotypic novelty (Stelkens, Schmid et al. 2009) both increase with genetic distance between the crossed species. The divergence time of the Congolese and Upper Nile lineage taxa, interpreted in the context of this previous experimental work, lets us predict that a hybrid swarm between these lineages would contain both intrinsic incompatibilities and transgressive trait variation.”

Comment 1.4: This is also wrapped up in the idea of admixture (usually thought to occur between populations) and hybridization (usually thought to occur between species). Obviously, when comparing vertebrate radiations with those of plants and arthropods, this distinction becomes very obscure (many vertebrate “species” would likely be called populations in plants or arthropods Though it may also happen the other way round, but the point is that there is a lot of wiggle room).

>> We agree with the reviewer that admixture is a term that is commonly used in population genetics even though it is also often used for genetic mixing between species (not only conspecific populations). However, we view the distinction between these terms differently. We view the term

“hybridization” as reproduction between members of divergent populations or species (a process), and “admixture” as the legacy of introgressive hybridization (a pattern). Note that admixture does not necessarily produce populations with mixed ancestry from both hybridizing taxa if there is strong extrinsic or intrinsic selection against hybrids. This view is consistent with the way we now use the terms throughout the paper and in line with definitions e.g. in Noor and Feder (2006); Buerkle and Lexer (2008); Abbott, Albach et al. (2013); Arnold (2015).

Comment 1.5: *Line 230 “Some of these sites that are fixed for alternative alleles in the parental lineages may be involved in negative epistatic interactions when brought together in the hybrid swarm” Note that this is a phenomenon that Carson (1990) alluded to.*

>> Carson, Lockwood et al. (1990) write about the idea of metapopulation structures and repeated founder events promoting the evolution of unique species citing Carson and Templeton (1984). They mention “polygenic recombinational genetic variability” which may counteract the loss of alleles in founder events citing Nei, Maruyama et al. (1975). As far as we understand, Carson et al. (1990) do not mention incompatibilities. Does the reviewer refer to a different paper? Nevertheless, we do now cite Carson et al., (1990) for the potentially constructive role of hybridization (see comment 1.6), and we were glad for the suggestion to carefully read this paper.

Comment 1.6: *Line 250 “However, finding evidence that hybridization between distantly related species may facilitate the onset of an adaptive radiation⁴ has been more challenging.” Hugely challenging, though widely discussed. Would be great to highlight the early discussions more, and that they basically reached stalemate because they couldn’t be tested.*

>> In the revised Introduction, we now discuss the ideas and evidence for a role of hybridization in adaptive radiation in more depth (see also response to comment 1.2). We now also give references to the sentence “Hybridization between species can instantaneously boost genetic variation” (lines 38-39), acknowledging earlier literature on the role of hybridization in diversification (Anderson and Stebbins 1954; Carson, Lockwood et al. 1990; Rieseberg 1997; Seehausen 2004; Mallet 2007; Abbott, Albach et al. 2013; Arnold 2015).

Comment 1.7: *Another point here is the “distantly related”. This relates to the point regarding the “relatively distantly related species” above, and also makes comparison between radiations of disparate taxa really difficult. The thing is, the species are only distantly related relative to the extremely closely related species swarm that comprises the radiation.*

>> We agree with the reviewer that “distant” can mean very different things in different contexts. In the revised manuscript, we reference previous experimental work that places the divergence time of the Congolese and Upper Nile lineages in an explicit context (see comment 1.3). In the revised Introduction paragraph about previous findings of hybrid origin of adaptive radiation, the word “distant” is not used anymore (see comment 1.2).

Comment 1.8: *Line 247 “Hybridization among members of adaptive radiations has been shown to facilitate additional speciation events in many systems including Darwin’s finches, Heliconius butterflies, Rhagoletis fruit flies, sunflowers, Lord Howe Coprosma ...”. These are very different systems, and only some would be considered adaptive radiations. Rather than simply list these examples, would be good to highlight exactly what elements of the phenomenon are shown in these different systems – what is the same and what different. Note that this does not require extended*

discussion of each system, but rather simply placing the different studies in the context of what they actually bring to the table for understanding the role of hybridization and admixture in adaptive radiation.

>> We think that the most important distinction is whether hybridization seeded the origin of an adaptive radiation (“hybrid swarm origin of adaptive radiation” of Seehausen 2004) or if hybridization among members of the adaptive radiation facilitated further speciation events (“syngameon hypothesis” of Seehausen 2004), as we discuss in response to comment 1.2. We have clarified this in the revised Introduction paragraphs (see response to comment 1.2).

We agree with the reviewer that for both scenarios, the evidence for different elements of these scenarios differs among cases of adaptive radiation supporting them. Following the suggestion by the reviewer, we now present evidence supporting the “syngameon hypothesis” or the “hybrid swarm origin of adaptive radiation” in two separate paragraphs in the Introduction and in each paragraph, we indicate if evidence for a diversifying role of hybridization has been found. We have also removed the *Helianthus* sunflower and *Coprosma* examples as they may not be considered adaptive radiations and added sailfin silversides as an additional example (see revised Introduction paragraphs given in the response to comment 1.2).

Comment 1.9: *Line 254 “Hybridization between distantly related species prior to the beginning of adaptive radiation has also been demonstrated in a plant radiation, the Hawaiian silverswords “ This is the first mention of the phenomenon being well documented in another radiation. But that was a long time ago, and there were major limitations in this study. The point here is that people have been looking for concrete evidence of the phenomenon for a very long time, and this study by Meier et al is perhaps the first to provide really solid data as to how it might happen, and under what circumstances.*

>> We hope that in the revised Introduction (see responses to comments 1.2, 1.6 and 1.8) and Discussion all of this does now become clearer. We now end the main text (Lines 303-310) as follows: “That hybridization may fuel entire adaptive radiations has been hypothesized for several systems but robust evidence was so far been confined to the allopolyploid Hawaiian silverswords (Barrier, Baldwin et al. 1999). Here, we report strong evidence that hybrid ancestry fuelled a large animal adaptive radiation. Thanks to the advent of high-throughput sequencing, our power to test for hybridization in the past and to study its impact on subsequent evolution has increased enormously. Future studies will hopefully reveal if hybridization in the ancestry of major adaptive radiations is more widespread and whether its occurrence may explain some of the observed large variation among lineages in the rates and volume of species diversification.”

Reviewer #2:

Comment 2.1: *This paper uses genomic data to argue that the Lake Victoria region superflock of haplochromine cichlids initiated as a result of a hybridization event between a species or species from the upper Nile valley, and others from the Congo region. In general I found this study convincing, but in places I found the reasoning and evidence frustratingly difficult to understand. Part of this is due to the brevity of the exposition, coupled with the complexity of multi-species, whole-genome evidence. In my view an article like this should steer away from discussions that require the reader de-coding obscure forms of explanation in figures that are there only in order to fit a lot of complexity into the slim format of a Nature-style article. For instance it took me a long time to work out what b, s, p, w, t, g were in Fig. 2b; I think at least some of these letters and their explanations should be in the legend, and perhaps in the text as examples, where I hunted for it,*

only to find it eventually in tiny parentheses in Fig. 2a.

>> We are pleased that reviewer #2 finds our study convincing and apologize that he/she found parts of the manuscript difficult to understand. We very much appreciate that the reviewer has pointed us to parts of the manuscript he/she found difficult to understand, and we have worked to clarify these points in the revised version of the manuscript (see responses to the more detailed comments below). We have added to the caption of Fig. 2 that the abbreviations in (b) are found in (a): "...each Eastern and Upper Nile taxon (P3) separately (abbreviations given in a)..." and we refer to them in the main text as suggested by the reviewer in comment 2.3 (see response to comment 2.3).

Comment 2.2: I'm unclear why the relatively weak mitochondrial information was deemed worth including, given that the vast weight of the phylogenetic evidence is from the genomic resequencing (RAD) data. Cutting this would save some space. The relatively short sequence lengths studied also would be prone to phylogenetic error. For similar reasons, the relatively weak evidence for time-dependent rates of molecular evolution seems a red herring here (lines 367-373), especially as one can't do anything about it in the dated tree. Much of the evidence may anyway be spurious due to difficulties with finding adequate priors for Bayesian phylogenetics, and the tendency for the tips of trees to be elongated in many programs. In other words it can be an artefact due to Bayesian mis-specification (see Brent Emerson 2007 Syst Biol. 56, 337–345).

>> The mitochondrial tree is only shown in the Supplementary Figures 1 (comparison to RADseq tree) and 3 (dated phylogeny), and not in the main text. We were thus not too worried about space limitations, but we do think that it provides complementary data to that of the RADseq tree, in that it underscores the large difference in the mtDNA versus nuclear history of the group, consistent with admixture between lineages being a repeated feature in cichlid evolution. We completely agree that the information in the RADseq tree is much stronger, hence our decision to only present the RADseq tree in the main text. However, the mitochondrial phylogeny of African cichlid radiations has shaped the literature about cichlid radiations for the past 20 years and the important nodes have always been (and are in our data too) strongly supported. It is the contrasting evolutionary history that we infer from our new genomic data versus the classical mitochondrial hypotheses that we want to point to by including the mitochondrial sequence analyses.

We also agree that the phylogenetic relationships among recent taxa and the length of the terminal branches are prone to bias and should thus not be over-interpreted. To prevent the readers from interpreting the recent time estimates, we overlaid grey rectangles over the time axes from 0-1 million years. We write in the caption of Supplementary Fig. 3: "As phylogenetic tree inference methods assumes a time-constant clock, whereas in fact the clock rate decays exponentially up to one million years (Ho and Larson 2006; Genner, Seehausen et al. 2007; Ho, Lanfear et al. 2011), ages of divergence events younger than 1 Ma are strongly overestimated and cannot be read from our graph (indicated by grey rectangles over the time scales)." The non-linear time-dependence in the first 1 million years is not a finding of our study, it is a problem discussed in earlier publications. We now refer to the review of Ho et al., 2011 in the first sentence discussing this issue to make this more clear (previously on line 367): "Estimating very recent divergence times with relaxed molecular clock approaches is challenging due to non-linearity of the clock rate in the very recent time range (reviewed in Ho, Lanfear et al. (2011)" (lines 413-415). We do not base any assumptions or time estimates on terminal branches. Our calibration nodes are relatively close in time (~1-9 million years ago) to the node that we are aiming to date with this tree (split of Congolese and Upper Nile lineages: 1.6-5.8 million years ago). They should thus be similarly affected by the biases on terminal branch lengths. We have now added the following sentence to the Methods paragraph: "However, given that the node we aim to date (Congo-Nilotic divergence time) is in the time range where the molecular clock is constant and its age is similar to the age of the

calibration nodes, we believe that this problem should not affect the estimate of the divergence time between the Congolese and the Upper Nile lineage” (lines 423-427). We agree that it is difficult to find adequate priors for the calibration nodes. This is why we present the results of different calibration sets with different priors, to show the large uncertainty in the time estimates.

Comment 2.3: l. 128: *here I'd mention that these two species are g and t in Fig. 2b.*

>> We thank the reviewer for this very helpful suggestion. We now write “We find evidence for gene flow from *H. gracilior* (g in Fig. 2b) and from *T. pharyngalis* (t in Fig. 2b) into each LVRS lake radiation” (lines 161-162).

Comment 2.4: *One problem with the analysis is that the RAD loci are largely anonymous, and so little information is gained as to the mechanism of speciation or the contributing genes to rapid radiation. To some extent a fuller discussion of the LWS opsin results would ameliorate this, but these data is treated in frustratingly little detail, and all of the functional information seems to be referring to other papers rather than being spelt out clearly here. We are more or less told the LWS opsin an important locus in ecology and evolution, and in adaptation to depth and turbidity, but the mechanisms are unclear. The mention in the results is brief, and there is nothing about it in the discussion, and the supplementary discussion does not add much. And yet a whole figure is devoted to this in the main text. I was left with many questions. Why would a shift in wavelength sensitivity be important in adaptation to differential exploitation of different light environments? How might this locus affect mate choice and sexual selection (as opposed to being told there are citations to data that it does affect it)? Are ALL the Nile species known (not just the two mentioned) fixed for the class II allele, and all the Congo species known to be fixed for the class I allele? If so, how do these ecologically important alleles fit their environments in the presumed ancestors? In other words, why might there be a difference in the first place? Adaptation? Historical accident? Why are the allelic classes referred to variously Class I vs. Class II (Figs. 4, S7 and supplementary discussion), H and L (Supplementary discussion), and Congolese and Upper Nile (supplementary discussion).*

>> We thank the reviewer for giving us the opportunity to expand our discussion of the relevance of the *LWS* opsin gene for the adaptive radiation and we hope that the revised paragraphs are now easier to understand without reading the cited literature. We have also realized that Supplementary Table 7 was missing, which shows the *LWS* opsin findings. We apologise for this omission. In the revised Results section, we now write (lines 178-188): “In addition, some of the allelic variation in the lake radiations has been shown to predate the origin of the LVRS (Terai, Seehausen et al. 2006; Brawand, Wagner et al. 2014), perhaps most notably that of the long-wavelength sensitive (*LWS*) opsin gene, one of the best-studied genes in Lake Victoria cichlids (Terai, Mayer et al. 2002). This gene plays a crucial role in adaptation to the steep ambient light gradients associated with water depth and turbidity gradients that are characteristic for the lakes in the Lake Victoria region. More red-shifted *LWS* opsin variants are beneficial in deep and murky water where the light spectrum is relatively more red-shifted due to particulate matter absorbing and scattering light of shorter wave lengths (Okullo, Ssenyonga et al. 2007; Seehausen, Terai et al. 2008). This opsin gene likely also plays a role in behavioural reproductive isolation because divergence in colour perception between species with different *LWS* opsin genotype (Maan, Hofker et al. 2006) is often associated with divergent male breeding colouration (Terai, Seehausen et al. 2006; Seehausen, Terai et al. 2008; Miyagi, Terai et al. 2012) which is an important mate choice cue (Selz, Pierotti et al. 2014).”

We have sequenced all individuals of the Congolese and Upper Nile lineage that we had access to, and these individuals represent all known species of these lineages except for one described species, *Thoracochromis petronius*, which may be part of the Upper Nile lineage but for which all DNA

extractions failed and no other samples are available. It is unclear why the Congolese and Upper Nile taxa have such functionally different haplotypes at the *LWS* opsin gene. One potential explanation may be that the Congolese lineage species are riverine and thus no deep water habitat is available, whereas the Upper Nile lineage species may have evolved in paleolake Obweruka (see Fig. S3) and survived in smaller intermittent paleolakes (precursors of Edward and Kivu), where the deep water habitat may have been similarly red-shifted as in modern Lake Victoria. However, these hypotheses are speculative and we prefer to not include them in the main text.

Regarding the Supplementary Discussion: Neither the “H and L haplotypes”, nor the “Congolese and Upper Nile variants” correspond to the major haplotype classes I and II. In the section where we use these terms, we discuss recombination between the ancestral haplotypes leading to additional recombinant haplotypes. The “H haplotype” belongs to haplotype class I, whereas the “L haplotype” is likely a recombinant between haplotype classes I and II. Congolese and Upper Nile variants refer to SNPs in the *LWS* opsin gene where the Congolese taxa and the Upper Nile taxa have different alleles. We thank the reviewer for pointing us to the unclear writing and the absence of Supplementary Table 7, which made it difficult to understand this Supplementary Discussion section. We have now revised the section “*LWS* opsin haplotypes” for improved clarity. As an example, we now write: “Whereas the clear-water H haplotype belongs to class I, the L haplotype seems to be a recombinant between the two haplotype classes (Supplementary Table 7).”

Comment 2.5: lines 230-233: "Some of these sites that are fixed for alternative alleles in the parental lineages may be involved in negative epistatic interactions when brought together in the hybrid swarm and differential sorting of such Bateson-Dobzhansky-Muller incompatibilities may facilitate reproductive isolation among the species emerging from the hybrid swarm." It's certainly possible but it is hardly a finding of this paper.

>> We agree that BDM incompatibilities are a hypothesis potentially explaining the observed enrichment of LV outliers among sites that are fixed for alternative alleles between the parental lineages. Following the comment by reviewer #1, we have now added a sentence stating that given the genetic divergence between the Congolese and Upper Nile lineages we would expect incompatibilities in hybrids of these lineages (based on results from experimental crosses, see comment 1.4). We hope that this also helps the reader to understand why we think that sites fixed for alternative alleles in the parental lineages may be involved in BDM incompatibilities in the hybrid population.

The evidence for BDM incompatibilities in our study stems from the comparison of ancestry category 4 (Congolese and Upper Nile lineage fixed for alternative alleles) to category 3 (only one allele found in Congolese taxa, second allele found in Upper Nile lineage taxa and thus only available in the LVRS due to hybridization). BDM incompatibilities would be fixed for alternative alleles in the parental lineages and would thus be assigned to category 4. Given that this category is strongly enriched for LV outliers (sites with high global F_{ST} among Lake Victoria species), some of these category 4 SNPs may indeed be BDM incompatibilities that were sorted differentially among the species that emerged from the hybrid swarm (see also our response to comment 2.9). If allelic variation introduced through hybridization was generally more often highly differentiated among Lake Victoria species, we would expect the same enrichment of LV outliers in category 3.

We have changed our wording in the Discussion to make this clearer: “Sites showing negative epistatic interactions (Bateson Dobzhansky Muller (BDM) incompatibilities) would be fixed for alternative alleles in the parental lineages and thus fall in ancestry category 4 in Fig. 5a. The strong enrichment of LV outliers in this category is in line with differential sorting of BDM incompatibilities among species that emerged from the hybrid swarm (Seehausen 2013; Hermansen, Haas et al. 2014; Schumer, Cui et al. 2015) which may have facilitated the origin of reproductive

isolation among the species arising in the adaptive radiation.” (lines 282-287).

Comment 2.6: lines 263-265: *"If hybridization prior to the onset of adaptive radiation is typical for particularly rapid and large species radiations, the importance of interspecific hybridization in the evolution of species diversity may have been underestimated". This is certainly true, but I was tempted to say, yes, but if it's not typical, then it may not have been underestimated! In other words, this is a rather empty statement with which to end a paper.*

>> In the revised Introduction we now discuss the evidence for a hybrid swarm origin in other adaptive radiations in more depth (see comment 1.2 above). This phenomenon is thought to be quite widespread and it may therefore be of general importance but this has been challenging to test. We have also rewritten the last paragraph which now states:

“That hybridization may fuel entire adaptive radiations has been hypothesized for several systems but robust evidence was so far been confined to the allopolyploid Hawaiian silverswords (Barrier, Baldwin et al. 1999). Here, we report strong evidence that hybrid ancestry fuelled a large animal adaptive radiation. Thanks to the advent of high-throughput sequencing, our power to test for hybridization in the past and to study its impact on subsequent evolution has increased enormously. Future studies will hopefully reveal if hybridization in the ancestry of major adaptive radiations is more widespread and whether its occurrence may explain some of the observed large variation among lineages in the rates and volume of species diversification.” (lines 303-310).

Comment 2.7: l. 299: *Why truncate at 84 nucleotides? I take it that the entire RAD sequence was used, and not just a single SNP from each restriction site? Why not paired end? And did you use each site for both forward and reverse directions of read from each restriction site after aligning to the reference, yielding a ~150 bp total read for each rad locus?*

>> The Illumina reads were originally 100 bp long. We removed the barcode sequences in all reads which are mostly 6 nt long resulting in reads of ~94 nt. We then trimmed the reads to 84 nt because the quality was low for the last 10 nt. To clarify this in the manuscript, we now write “The length of 84 nt results from removing the barcode (mostly 6 nt long) and trimming off the last 10 nt due to low quality at the read ends” (lines 346-348). Indeed, we used sequenced RAD loci in both directions from the RAD site. We did not use paired-end because it is twice as costly as sequencing single-end but does not produce the double amount of data. This is because in RAD libraries (in contrast to ddRAD libraries), one end starts with the restriction site (where *Sbf*I cuts) and the other end is randomly sheared as illustrated below. In the size selection step, we extract DNA fragments that are 300 to 600 bp long. Therefore, the second end of the RAD fragment, where Read 2 would start (shown in blue below), is at any position 300-600 bp away from the restriction site. If we had sequenced the second end as well (paired-end sequencing), we would get low depth of coverage for the second end because these reads would not fully overlap (illustrated below).

Comment 2.8: lines 481 to 505. *This method seems to cover only a few of the "categories" of ancestral origin of fig. 5 (i.e. categories 2-4). If possible, can some of these categories be lumped to*

make the whole easier to understand?

>> We thank the reviewer for pointing out that the categories 1, 5 and 6 were not explicitly stated in the Methods section (only in the figure caption). We now refer to all ancestry categories in the Methods section. We have removed category 6 which was the combination of categories 2 and 3. We agree that this improves the clarity of the figure. We are now directly comparing category 4 (sites fixed for alternative alleles in the Congolese and Upper Nile taxa) with category 2 (polymorphic in Congolese taxa and thus available in LVRS without hybridization) and with category 3 (second allele only found in Upper Nile lineage taxa and thus only available in the LVRS due to hybridization).

Comment 2.9: Fig. 5: I don't fully understand this figure. Delete Bateson-Dobzhansky-Muller incompatibilities, for which there is no evidence given, from the legend. From the description, I would have thought category 6 should be a bigger category than category 5, not smaller as shown. Overall I was confused by these categories and what they're supposed to show us; can they be reduced to a more manageable number. I would have thought that the "neutral" expectation that the allele frequency was approx. equal (4-18%) to the estimated ancestral fraction in the ancestor is prone to all sorts of other things like drift, and so I think that "polymorphic" would be a better overall category. Finally, are there really only 20 sites fixed for alternative alleles between the Congolese and Upper Nile species as in Fig. 5b?

>> The original category 6 (1991 SNPs total) was indeed much larger than category 5 (179 SNPs total) as the reviewer would have expected. The height of each bar corresponds to the proportion of LV outliers (sites that are outliers of high global F_{ST} among the 6 Lake Victoria species) among all SNPs in that category. The total number of SNPs in each category is given as number above the

category. We have revised the legend to make this clearer, which now states: “We assigned SNPs to five different ancestry categories according to the presence or absence of the two alleles in the Congolese and Upper Nile lineage taxa. The grey bars show the proportion of LV outliers among all SNPs in each ancestry category. Total SNP counts in each category and p-values of two-sided Fisher’s exact tests are shown on top.” Note, that we have now removed category 6 following the suggestion in comment 2.8. We have also improved the labels of the categories and revised the associated text in the Methods, Results and figure caption to improve clarity.

The much higher proportion of LV outliers in category 4 (Congolese and Upper Nile lineages fixed for alternative alleles) than in category 3 (monomorphic in Congolese and polymorphic in Upper Nile lineage taxa) suggests that sites fixed for alternative alleles in the parental lineages are particularly prone to be differentiated among species that emerged from the hybrid swarm. Given the divergence time between the Congolese and Upper Nile lineage, we would expect BDM incompatibilities in hybrids between the Congolese and Upper Nile lineage (see also response to comment 1.3). Loci with negative epistatic interactions (BDM incompatibilities) would all be assigned to category 4 as they would be fixed for different alleles in the two parental lineages. The enrichment of LV outliers in category 4 may thus be due to differential sorting of compatible combinations of alleles as suggested by Seehausen (2013); (Hermansen, Haas et al. 2014); Schumer, Cui et al. (2015). We have revised this section in the manuscript which now states as follows (lines 273-287): “Experimental crosses of cichlids have shown that intrinsic incompatibilities (Stelkens, Young et al. 2010; Stelkens, Schmid et al. 2015) and phenotypic novelty (Stelkens, Schmid et al. 2009) both increase with genetic distance between the crossed species. The divergence time of the Congolese and Upper Nile lineage taxa, interpreted in the context of this previous experimental work, lets us predict that a hybrid swarm between these lineages would contain both intrinsic incompatibilities and transgressive trait variation. In agreement with this prediction, we found that genomic sites with alternative alleles fixed in the two parental lineages are enriched for candidate loci of reproductive isolation, divergent adaptation and species differentiation in Lake Victoria (Fig. 5). This indicates that speciation was commonly associated with sorting of alleles brought together in the radiation ancestor by the admixture event. Sites showing negative epistatic interactions (Bateson Dobzhansky Muller (BDM) incompatibilities) would be fixed for alternative alleles in the parental lineages and thus fall in ancestry category 4 in Fig. 5a. The strong enrichment of LV outliers in this category is in line with differential sorting of BDM incompatibilities among species that emerged from the hybrid swarm (Seehausen 2013; Hermansen, Haas et al. 2014; Schumer, Cui et al. 2015) which may have facilitated the origin of reproductive isolation among the species arising in the adaptive radiation.”

We are now comparing category 4 (parental lineages fixed for alternative alleles) with categories 2 and 3 (polymorphic in the parental lineages) as the reviewer suggested. We do believe, though, that a comparison of category 4 with category 5 (initial maf 14-18%) is useful to test if the enrichment of LV outliers in category 4 is simply due to moderately high initial allele frequency. We agree that category 5 is influenced by drift. However, the enrichment is also significant if category 4 is compared against datasets with estimated initial allele frequencies ranging from 10 to 30% which contain LV outlier proportions of 8-12% (Supplementary Discussion).

There are only 20 sites fixed for alternative alleles in the Congolese and Upper Nile lineage taxa which are also outliers of high F_{ST} between the six Lake Victoria species. However, there are in total 100 sites fixed for alternative alleles in the Congolese and Upper Nile lineage taxa that passed all of our filters (e.g. missing data, variable in the six Lake Victoria species).

References cited in the point-by-point response

- Abbott, R., D. Albach, et al. (2013). "Hybridization and speciation." *J Evol Biol* **26**(2): 229-246.
- Anderson, E. and G. L. Stebbins (1954). "Hybridization as an Evolutionary Stimulus." *Evolution* **8**(4): 378-388.
- Arnold, M. L. (2015). *Divergence with genetic exchange*, Oxford University Press.
- Baldwin, B. G. and W. L. Wagner (2010). "Hawaiian angiosperm radiations of North American origin." *Annals of Botany*: mcq052.
- Barrier, M., B. G. Baldwin, et al. (1999). "Interspecific hybrid ancestry of a plant adaptive radiation: allopolyploidy of the Hawaiian silversword alliance (Asteraceae) inferred from floral homeotic gene duplication." *Mol Biol Evol* **16**.
- Brawand, D., C. E. Wagner, et al. (2014). "The genomic substrate for adaptive radiation in African cichlid fish." *Nature* **513**(7518): 375-381.
- Buerkle, C. A. and C. Lexer (2008). "Admixture as the basis for genetic mapping." *Trends in Ecology & Evolution* **23**(12): 686-694.
- Carlquist, S. (1974). "Island biology." *Columbia University Press: New York & London*. 660pp **581**: 5279.
- Carson, H. L., J. P. Lockwood, et al. (1990). "Extinction and recolonization of local populations on a growing shield volcano." *Proc Natl Acad Sci USA* **87**(18): 7055-7057.
- Carson, H. L. and A. R. Templeton (1984). "Genetic revolutions in relation to speciation phenomena: the founding of new populations." *Annual review of ecology and systematics*: 97-131.
- Feder, J. L., X. Xie, et al. (2005). "Mayr, Dobzhansky, and Bush and the complexities of sympatric speciation in *Rhagoletis*." *Proc Natl Acad Sci USA* **102**(suppl 1): 6573-6580.
- Genner, M. J., O. Seehausen, et al. (2007). "Age of cichlids: New dates for ancient lake fish radiations." *Mol Biol Evol* **24**(5): 1269-1282.
- Genner, M. J. and G. F. Turner (2012). "Ancient hybridization and phenotypic novelty within Lake Malawi's cichlid fish radiation." *Mol Biol Evol* **29**.
- Gilbert, L. (2003). "Adaptive novelty through introgression in *Heliconius* wing patterns: evidence for shared genetic "tool box" from synthetic hybrid zones and a theory of diversification." *Ecology and Evolution Taking Flight: Butterflies as Model Systems*: 281-318.
- Herder, F., A. W. Nolte, et al. (2006). "Adaptive radiation and hybridization in Wallace's Dreamponds: evidence from sailfin silversides in the Malili Lakes of Sulawesi." *Proc Biol Sci B* **273**(1598): 2209-2217.
- Hermansen, J. S., F. Haas, et al. (2014). "Hybrid speciation through sorting of parental incompatibilities in Italian sparrows." *Mol Ecol* **23**(23): 5831-5842.
- Ho, S. Y., R. Lanfear, et al. (2011). "Time-dependent rates of molecular evolution." *Mol Ecol* **20**(15): 3087-3101.
- Ho, S. Y. and G. Larson (2006). "Molecular clocks: when times are a-changin'." *Trends Genet* **22**(2): 79-83.
- Hudson, A. G., P. Vonlanthen, et al. (2011). "Rapid parallel adaptive radiations from a single hybridogenic ancestral population." *Proc R Soc B* **278**(1702): 58-66.
- Joyce, D. A., D. H. Lunt, et al. (2011). "Repeated colonization and hybridization in Lake Malawi cichlids." *Current Biol* **21**(3): R108-R109.
- Keller, I., C. E. Wagner, et al. (2013). "Population genomic signatures of divergent adaptation, gene flow and hybrid speciation in the rapid radiation of Lake Victoria cichlid fishes." *Mol Ecol* **22**(11): 2848-2863.
- Lamichhaney, S., J. Berglund, et al. (2015). "Evolution of Darwin's finches and their beaks revealed by genome sequencing." *Nature* **518**(7539): 371-375.
- Lindqvist, C. and V. A. Albert (2002). "Origin of the Hawaiian endemic mints within North American *Stachys* (Lamiaceae)." *American Journal of Botany* **89**(10): 1709-1724.
- Lindqvist, C., T. J. Motley, et al. (2003). "Cladogenesis and reticulation in the Hawaiian endemic

- mints (Lamiaceae)." Cladistics **19**(6): 480-495.
- Lynch, M. and J. S. Conery (2000). "The evolutionary fate and consequences of duplicate genes." Science **290**(5494): 1151-1155.
- Maan, M. E., K. D. Hofker, et al. (2006). "Sensory drive in cichlid speciation." Am Nat **167**(6): 947-954.
- Mallet, J. (2007). "Hybrid speciation." Nature **446**(7133): 279-283.
- Meier, J. I., V. C. Sousa, et al. (2016). "Demographic modelling with whole-genome data reveals parallel origin of similar *Pundamilia* cichlid species after hybridization." Mol Ecol.
- Meyer, B. S., M. Matschiner, et al. (2016). "Disentangling incomplete lineage sorting and introgression to refine species-tree estimates for Lake Tanganyika cichlid fishes." Systematic Biology.
- Miyagi, R., Y. Terai, et al. (2012). "Correlation between nuptial colors and visual sensitivities tuned by opsins leads to species richness in sympatric Lake Victoria cichlid fishes." Mol Biol Evol **29**(11): 3281-3296.
- Nei, M., T. Maruyama, et al. (1975). "The bottleneck effect and genetic variability in populations." Evolution: 1-10.
- Noor, M. A. F. and J. L. Feder (2006). "Speciation genetics: evolving approaches." Nature Reviews Genetics **7**(11): 851-861.
- Okullo, W., T. Ssenyonga, et al. (2007). "Parameterization of the inherent optical properties of Murchison Bay, Lake Victoria." Applied Optics **46**(36): 8553-8561.
- Pardo-Diaz, C., C. Salazar, et al. (2012). "Adaptive introgression across species boundaries in *Heliconius* butterflies." PLoS Genet **8**(6): e1002752.
- Rieseberg, L. H. (1997). "Hybrid origins of plant species." Ann Rev Ecol Syst: 359-389.
- Salzburger, W., S. Baric, et al. (2002). "Speciation via introgressive hybridization in East African cichlids?" Mol Ecol **11**.
- Schliewen, U. K. and B. Klee (2004). "Reticulate sympatric speciation in Cameroonian crater lake cichlids." Front Zool **1**(5): issue 5.
- Schumer, M., R. Cui, et al. (2015). "Reproductive isolation of hybrid populations driven by genetic incompatibilities." PLoS Genet **11**(3): e1005041.
- Seehausen, O. (2004). "Hybridization and adaptive radiation." Trends Ecol Evol **19**(4): 198-207.
- Seehausen, O. (2013). "Conditions when hybridization might predispose populations for adaptive radiation." J Evol Biol **26**(2): 279-281.
- Seehausen, O., Y. Terai, et al. (2008). "Speciation through sensory drive in cichlid fish." Nature **455**(7213): 620-U623.
- Selz, O. M., M. E. R. Pierotti, et al. (2014). "Female preference for male color is necessary and sufficient for assortative mating in 2 cichlid sister species." Behav Ecol **25**(3): 612-626.
- Selz, O. M., R. Thommen, et al. (2014). "Behavioural isolation may facilitate homoploid hybrid speciation in cichlid fish." J Evol Biol **27**(2): 275-289.
- Stelkens, R. B., C. Schmid, et al. (2015). "Hybrid breakdown in cichlid fish." Plos One **10**(5): e0127207.
- Stelkens, R. B., C. Schmid, et al. (2009). "Phenotypic novelty in experimental hybrids is predicted by the genetic distance between species of cichlid fish." Bmc Evol Biol **9**.
- Stelkens, R. B. and O. Seehausen (2009). "Phenotypic divergence but not genetic distance predicts assortative mating among species of a cichlid fish radiation." Journal of Evolutionary Biology **22**(8): 1679-1694.
- Stelkens, R. B., K. A. Young, et al. (2010). "The Accumulation of Reproductive Incompatibilities in African Cichlid Fish." Evolution **64**(3): 617-632.
- Terai, Y., W. E. Mayer, et al. (2002). "The effect of selection on a long wavelength-sensitive (LWS) opsin gene of Lake Victoria cichlid fishes." Proc Natl Acad Sci U S A **99**(24): 15501-15506.
- Terai, Y., O. Seehausen, et al. (2006). "Divergent selection on opsins drives incipient speciation in Lake Victoria cichlids." Plos Biol **4**(12): 2244-2251.

- The Heliconius Genome Consortium (2012). "Butterfly genome reveals promiscuous exchange of mimicry adaptations among species." Nature **487**(7405): 94-98.
- Wallbank, R. W., S. W. Baxter, et al. (2016). "Evolutionary novelty in a butterfly wing pattern through enhancer shuffling." PLoS Biol **14**(1): e1002353.
- Weiss, J. D., F. P. Cotterill, et al. (2015). "Lake Tanganyika—A 'Melting Pot' of Ancient and Young Cichlid Lineages (Teleostei: Cichlidae)?" PloS one **10**(4): e0125043.

REVIEWERS' COMMENTS:

Reviewer #1 (Remarks to the Author):

The authors have done an excellent and thorough job of addressing previous concerns/ comments I had. For the comment on Carson et al: The main point with his work is that he wrote quite a bit about epistatic genetic variance and that the "balanced condition which is so important in the adjustment of phenotypes to the environment, can be perturbed and some of it converted to additive variance under the conditions existing in small populations" (from *Geojournal* 1992, but there are a number of others, including the *Genetics of Natural Populations*, 1995). Carson was somewhat ahead of his time in coming up with these ideas - although he did generate extraordinary amounts of ground-breaking data on relationships between *Drosophila* based on giant salivary chromosomes.

I think that the manuscript now reads very well, and places the work sufficiently within a broader context.

Reviewer #2 (Remarks to the Author):

The authors have accepted some of my recommendations, and rejected others, as expected. I've re-read the paper, which I already had found quite convincing, as stated earlier.

I still don't believe the authors need to raise the spectre of increased rates of molecular evolution near the present here, especially as they were unable to do anything about it. It would save space to avoid this discussion. However, I see now that some of the authors have a stake in the hypothesis (which I regard as still very much unproved and speculative), so I do at least understand why it's there.

There may be well be an old hypothesis based on mtDNA fragments, but that's really no longer of interest when we have complete genome data. I suggest that the whole issue could be dealt with by a couple of sentences mentioning that the older low-resolution mtDNA fragment data conflicts with nuclear data, which is also expected based on hybridization.

"The closest relatives were inferred with maximum likelihood trees based on these data and on mitochondrial sequences of the same fish." How about omitting the mitochondrial sequences as evidence, given the much greater information used from the RAD data?

For similar reasons, I'd say the mitochondrial clock based on mitochondrial fragments is also highly suspect. It's the clock of the mitochondrial DNA divergences, not necessarily anything to do with times referring to speciation or species divergence. As is well known, a species tree and speciation clock based on a single non-recombining locus is highly misleading because of incomplete lineage sorting in a rapidly speciating lineage like this, and also because of the potential for gene flow between species, also in lineages like this one. The short regions sequenced are anyway liable to high rates of phylogenetic and clock

estimation error. At the very least, the authors should mention these problems if they insist on using these data in this way. Yes, it's sad, but much earlier work based on mtDNA genes needs to be revisited. Sorry. But it was the same for allozymes as well.

In spite of what the authors say in their response (that Congolese forms mainly come from shallow water), I found it amusing because the Congo River is renowned to be the deepest river on the planet, having some waters in the Central Congo over 220m in depth, completely dark, and in particular that have blind white fish that die after being brought to the surface. It's possible that the authors mean that the Eastern, upper Congo, which contain fish that would have seeded newly arising Great Lakes of Africa are all shallow waters.

See: <http://www.amnh.org/explore/science-bulletins/bio/documentaries/evolution-in-action-isolation-and-speciation-in-the-lower-congo-river/article-fish-evolution-takes-a-wild-ride/>

The authors have sorted out what was a rather vapid last sentence. I'd try to go even more punchy, and suggest further editing such as:

"Future studies will [delete: hopefully] reveal if hybridization in the ancestry of [other] major adaptive radiations is [delete: more] widespread and whether its occurrence may explain [the great] variation among lineages in [delete: the] rates [delete: and volume] of species diversification."

Point-by-point response

Reviewer #1:

The authors have done an excellent and thorough job of addressing previous concerns/ comments I had. For the comment on Carson et al: The main point with his work is that he wrote quite a bit about epistatic genetic variance and that the “balanced condition which is so important in the adjustment of phenotypes to the environment, can be perturbed and some of it converted to additive variance under the conditions existing in small populations” (from Geojournal 1992, but there are a number of others, including the Genetics of Natural Populations, 1995). Carson was somewhat ahead of his time in coming up with these ideas - although he did generate extraordinary amounts of ground-breaking data on relationships between Drosophila based on giant salivary chromosomes.

I think that the manuscript now reads very well, and places the work sufficiently within a broader context.

>> We thank the reviewer for the clarification and are happy that he/she is satisfied with our revisions and responses.

Reviewer #2:

2.1 *The authors have accepted some of my recommendations, and rejected others, as expected. I've re-read the paper, which I already had found quite convincing, as stated earlier.*

>> We thank the reviewer for rereading the manuscript and we appreciate that he/she still finds it convincing.

2.2 *I still don't believe the authors need to raise the spectre of increased rates of molecular evolution near the present here, especially as they were unable to do anything about it. It would save space to avoid this discussion. However, I see now that some of the authors have a stake in the hypothesis (which I regard as still very much unproved and speculative), so I do at least understand why it's there.*

There may be well be an old hypothesis based on mtDNA fragments, but that's really no longer of interest when we have complete genome data. I suggest that the whole issue could be dealt with by a couple of sentences mentioning that the older low-resolution mtDNA fragment data conflicts with nuclear data, which is also expected based on hybridization.

"The closest relatives were inferred with maximum likelihood trees based on these data and on mitochondrial sequences of the same fish." How about omitting the mitochondrial sequences as evidence, given the much greater information used from the RAD data?

For similar reasons, I'd say the mitochondrial clock based on mitochondrial fragments is also highly suspect. It's the clock of the mitochondrial DNA divergences, not necessarily anything to do with times referring to speciation or species divergence. As is well known, a species tree and speciation clock based on a single non-recombining locus is highly misleading because of incomplete lineage sorting in a rapidly speciating lineage like this, and also because of the

potential for gene flow between species, also in lineages like this one. The short regions sequenced are anyway liable to high rates of phylogenetic and clock estimation error. At the very least, the authors should mention these problems if they insist on using these data in this way. Yes, it's sad, but much earlier work based on mtDNA genes needs to be revisited. Sorry. But it was the same for allozymes as well.

>> We agree with the reviewer that our RADseq and whole genome data is much more powerful than the mitochondrial sequence dataset. This is why we base all our analyses on the NGS data (except for the divergence time estimations) and why we show the mitochondrial data only in the Supplementary Information. We have now also moved most of the explanations on how we inferred the mitochondrial chronogram to the Supplementary Methods to give this part less weight and to follow the suggestion of the editor to move Methods to Supplementary Information to reduce the number of references. We now only write in the Methods section of the main text: “Dated phylogenies were reconstructed based on mitochondrial D-loop (Kocher *et al.*, 1989) and ND2 (Kocher *et al.*, 1995) sequences using BEAST v. 2.3.0 (Bouckaert *et al.*, 2014) and four different sets of calibration nodes (see Supplementary Methods). We caution that the mitochondrial tree only shows the phylogeny of the maternal line and that time estimates more recent than one million years are most likely overestimates because of the increase of molecular rates towards the recent (see Supplementary Methods).”

We believe that it is important to show the incongruences between mtDNA and nuclear trees and to show the mtDNA tree in the supplementary is necessary to allow direct comparisons with the rich published data on cichlid fish phylogeography, which is nearly all based on mtDNA. To compare our nuclear RADseq data to the published older data and place our data into the context of previous knowledge, we need to provide the mitochondrial sequences, and the relationships among them, for our samples. Without this our new data would be incomparable to previous data and would effectively reside in a vacuum.

We agree with the reviewer that “*earlier work based on mtDNA genes needs to be revisited*” and by showing the mtDNA tree in the supplementary figure side-by-side with the RADseq tree, we highlight this point.

2.3 *In spite of what the authors say in their response (that Congolese forms mainly come from shallow water), I found it amusing because the Congo River is renowned to be the deepest river on the planet, having some waters in the Central Congo over 220m in depth, completely dark, and in particular that have blind white fish that die after being brought to the surface. It's possible that the authors mean that the Eastern, upper Congo, which contain fish that would have seeded newly arising Great Lakes of Africa are all shallow waters.*

See: <http://www.amnh.org/explore/science-bulletins/bio/documentaries/evolution-in-action-isolation-and-speciation-in-the-lower-congo-river/article-fish-evolution-takes-a-wild-ride/>

>> Here, the reviewer refers to our hypothesis mentioned in the previous point-by-point response (not in the manuscript) that there may be an association of the LWS opsin haplotypes derived from the Congolese or Upper Nile lineages with the rather shallow riverine or deep (paleo)lake environments, respectively. It is correct that the lower parts of the Congo River (below Malebo Pool) are very deep. However, this is hundreds of kilometres away from the Lake Victoria region and is a region that is ichthyogeographically strongly isolated from the upper Congo by many kilometres of exceptionally steep rapids. The Congolese *Astatotilapia* relatives of the LVRS occur in shallow tributaries of the upper Congo River and not in the deep lower Congo.

2.4 *The authors have sorted out what was a rather vapid last sentence. I'd try to go even more punchy, and suggest further editing such as:*

"Future studies will [delete: hopefully] reveal if hybridization in the ancestry of [other] major adaptive radiations is [delete: more] widespread and whether its occurrence may explain [the great] variation among lineages in [delete: the] rates [delete: and volume] of species diversification."

>> We have changed the last sentence following suggestions of the reviewer. We have not removed "and volume" because we think that not just the rate of diversification but also the number of species produced may be influenced by hybridization. We now write: "Future studies will reveal if hybridization in the ancestry of major adaptive radiations is widespread, and whether its occurrence may explain some of the observed large variation in the rates and volume of species diversification among lineages."